# CFPO: Counterfactual Policy Optimization for Multimodal Reasoning

**Zhangyuan Yu** [* 1]  **Wanran Sun** [* 1]  **Guangjing Yang** [1]  **Xiaohu Wu** [1]  **Qicheng Lao** [1 †]

## Abstract

Large Vision-Language Models (LVLMs) have demonstrated remarkable capabilities in multimodal reasoning. However, prevailing reinforcement learning (RL) paradigms lack explicit counterfactual enhancement and causal learning mechanisms. This fundamental deficiency results in severe grounding failures, manifesting as a tendency to ignore visual evidence in favor of language priors or exhibiting hallucination drift during long chain-of-thought reasoning. To address this root cause, we propose *CounterFactual Policy Optimization* (**CFPO**), a novel framework that enforces causal consistency between visual perception and textual reasoning. CFPO introduces a cross-modal counterfactual enhancement mechanism, which regularizes the policy by maximizing the discrepancy between the model's predictions and those from a counterfactual state where critical visual cues are suppressed. This approach seamlessly integrates with standard algorithms like GRPO and DAPO without requiring external reward models or additional supervision. Extensive experiments demonstrate that CFPO significantly improves reasoning fidelity, achieving consistent gains of 3.17%-6.25% over standard RL baselines and 1.32%-2.13% over the state-of-the-art perception-aware method (PAPO). Code is available at github.

## 1. Introduction

Large Vision-Language Models (LVLMs) have recently demonstrated remarkable progress in multimodal reasoning, visual question answering, and complex instruction following (Bai et al., 2025b; Li et al., 2024; Chen et al., 2024a;b;c). Despite these advances, a fundamental misalignment persists in how these models are optimized via reinforcement learning (RL). Prevailing RL paradigms—such as PPO and GRPO—predominantly optimize for outcome correctness, rewarding models solely based on the final answer (Schulman et al., 2017; Shao et al., 2024; Yu et al., 2025). From a causal perspective, such outcome-driven objectives do not explicitly enforce **cross-modal causal consistency** in the reasoning process. As a result, the optimizer cannot reliably distinguish whether a correct response stems from genuine cross-modal understanding or from exploiting **spurious correlations** inherent in the training data.

In the absence of causal constraints, models are susceptible to **shortcut learning** in multi-modal scenarios, bypassing robust reasoning in favor of statistical heuristics (Geirhos et al., 2020). We identify a spectrum of causal inconsistencies in LVLMs, governed by the pathological distribution of cross-modal saliency: (1) **Cross-Modal Saliency Deficiency:** The model ignores visual evidence and relies on the parametric knowledge of the LLM backbone to "guess" the answer. This phenomenon is deeply rooted in the dominance of language priors (Niu et al., 2021), often manifesting as the hallucination of non-existent visual entities (e.g., imaginary geometric segments) to force-fit a legitimate-looking but groundless reasoning chain. (2) **Cross-Modal Saliency Misalignment:** The model attends to perceptually salient but semantically irrelevant visual cues (Li et al., 2023b; Zhou et al., 2024). Furthermore, the model often suffers from coarse grounding in correctly located regions, where it fails to precisely align high-level concepts with fine-grained pixels (e.g., specific characters), establishing a false causal link between fuzzy visual features and hallucinated text. (3) **Cross-Modal Saliency Inertia:** A subtle yet critical failure where the visual evidence is correctly attended to but becomes an immutable anchor. Here, the strong coupling between visual facts and text prevents the model from updating its reasoning in response to hypothetical instructions (e.g., "what if..."), permitting static visual facts to override logical causal interventions. These phenomena collectively represent a failure in causal grounding: the generated reasoning path is not causally sensitive to the critical visual evidence required by the task. More details will be discussed in Section 5.2.

To mitigate such pathological distribution, recent works have explored counterfactual approaches. Inference-time strategies, such as VCD (Leng et al., 2024), M3ID (Favero

---

[*]Equal contribution [1]Beijing University of Posts and Telecommunications, Beijing, China. Correspondence to: Qicheng Lao <qicheng.lao@bupt.edu.cn>.

*Proceedings of the 43rd International Conference on Machine Learning*, Seoul, South Korea. PMLR 306, 2026. Copyright 2026 by the author(s).

et al., 2024), and ICD (Wang et al., 2024), construct counterfactual baselines by adding visual noise or misleading instructions to calibrate output distributions. However, these decoding-time interventions incur double computational overhead and fail to fundamentally rectify the model's internal parameters. In the training domain, data-driven alignment approaches like DeFacto (Xu et al., 2025) and CF-VLM (Zhang et al., 2025) enforces grounding via curated counterfactual datasets, yet it relies on labor-intensive supervision and AI-generated content rather than autonomous reinforcement learning. While PAPO (Wang et al., 2025b) attempts to integrate perception-aware constraints into RL via random input masking, this coarse intervention at the input pixel level often fails to precisely isolate the fine-grained semantic features driving the high-level reasoning.

Consequently, a robust solution must internalize causal verification directly into the optimization loop, avoiding both inference latency and coarse approximations. Addressing this challenge requires moving beyond standard associative learning to a mechanism that explicitly verifies the necessity of the attended information at the representation level. To this end, we propose **Counterfactual Policy Optimization** (CFPO), a novel training framework that enforces causal consistency by integrating **Cross-Modal Counterfactual Intervention** into the policy update loop. The central idea of CFPO is to perform a causal necessity test: we construct a **Counterfactual Path** within the multi-head attention layers to quantify how the policy distribution shifts when specific cross-modal **High Saliency Regions** are causally suppressed. This intervention serves a dual purpose: it penalizes *Cross-Modal Saliency Deficiency* and *Cross-Modal Saliency Misalignment* by forcing the model to attend to semantically relevant visual cues, while simultaneously breaking *Cross-Modal Saliency Inertia* by disentangling perception from inference. By learning to distinguish between the presence and absence of visual evidence, the model gains the flexibility to execute logical interventions during reasoning.

We integrate this mechanism into Group Relative Policy Optimization (GRPO) (Shao et al., 2024) and Decoupled Clip and Dynamic Sampling Policy Optimization (DAPO) (Yu et al., 2025) via a **Counterfactual Regularization** term. This objective compels the model to maximize the **Effective Causal Contribution**, ensuring that the generation is sensitive to the presence of critical visual cues. Unlike prior perception-aware methods that implicitly align modalities, CFPO provides a rigorous causal guarantee that the reasoning process is grounded in valid evidence.

Extensive experiments on multimodal reasoning benchmarks demonstrate that CFPO significantly improves performance, particularly in tasks requiring rigorous logic, such as mathematical reasoning and complex Chain-of-Thought (CoT) generation. By suppressing shortcut learning, CFPO not only reduces hallucinations but also enhances the robustness and interpretability of LVLMs, offering a principled path toward causally consistent multimodal intelligence.

**Conflict of Interest Disclosure** All authors disclosed no relevant relationships.

## 2. Preliminaries

### 2.1. Group Relative Policy Optimization

Group Relative Policy Optimization (GRPO) (Shao et al., 2024) streamlines PPO (Schulman et al., 2017) by removing the value model and estimating advantages via group-based sampling. Given a multimodal dataset containing visual inputs $I$, text queries $q$, and ground truth answers $\alpha$, the GRPO objective for the rollout policy $\pi_\theta$ is defined as:

$$\mathcal{J}_{GRPO}(\theta) = \mathbb{E}_{[\{o_i\}_{i=1}^G \sim \pi_{\theta_{old}}(O|q,I)]} \frac{1}{G} \sum_{i=1}^{G} \left( \hat{J}_{clip} - \beta KL_{ref} \right),$$

$$\hat{J}_{clip} = \min \left( r_i(\theta) A_i, \text{clip} \left( r_i(\theta), 1 - \epsilon_l, 1 + \epsilon_h \right) A_i \right),$$
(1)

where $r_i(\theta) = \frac{\pi_\theta(o_i|q,I)}{\pi_{\theta_{old}}(o_i|q,I)}$ is the probability ratio, and $G$ denotes the group size of outputs $O$ sampled from $\pi_{\theta_{old}}$. $KL_{ref} = \mathbb{D}_{KL}(\pi_\theta||\pi_{ref})$ is the regularization term used to prevent excessive deviation from the reference policy $\pi_{ref}$, controlled by coefficient $\beta$. Following (Zheng et al., 2025), we adopt the clip-higher configuration ($\epsilon_l = 0.2, \epsilon_h = 0.3$) for $\hat{J}_{clip}$. The advantage $A_i$ is computed by normalizing the rewards derived from a rule-based answer verifier $AV$ (Zheng et al., 2025):

$$A_i = \frac{R_i - \text{mean}(\{R_1, ..., R_G\})}{\text{std}(\{R_1, ..., R_G\})},$$
(2)

where the reward is set as $R_i = 1.0$ if $AV(\alpha, o_i)$ else $0.0$.

### 2.2. Decoupled Clip and Dynamic Sampling Policy Optimization

As a representative evolution of the GRPO framework, Decoupled Clip and Dynamic Sampling Policy Optimization (DAPO) (Yu et al., 2025) further refines the update stability by introducing mechanisms such as Clip-Higher, Dynamic Sampling, and Token-Level Policy Gradient Loss. DAPO removes the reference KL penalty (i.e., $\beta = 0$) and introduces a Decoupled Clipping mechanism with distinct bounds to encourage broader exploration. The dynamic sampling enforces the constraint: $0 < |\{o_i \mid \text{is\_equivalent}(\alpha, o_i)\}| < G$, which prevents invalid gradient estimation where groups with all correct/incorrect responses yield zero variance in advantage.

In our work, the proposed CFPO framework is integrated and validated on both GRPO and DAPO.

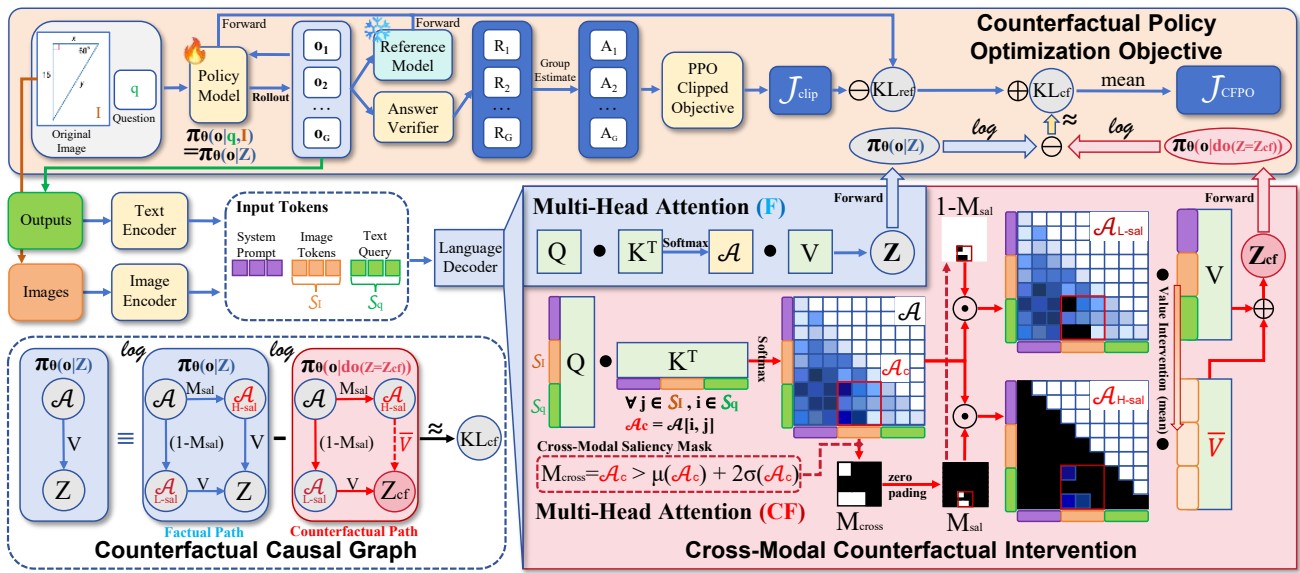

*Figure 1.* Overview of CFPO. We introduce a **Counterfactual Path** (Red) alongside the standard **Factual Path** (Blue). The core intervention occurs at the attention output level: $Z$ represents the original feature representation, whereas $Z_{cf}$ is the counterfactual representation with high-saliency visual cues suppressed by the mask $M_{sal}$. By maximizing the divergence ($KL_{cf}$) between predictions derived from $Z$ and $Z_{cf}$, CFPO forces the policy to ground its reasoning in valid visual evidence.

## 3. Methods

We propose **Counterfactual Policy Optimization** (CFPO), a principled framework designed to rectify the pathological cross-modal saliency distributions by strictly enforcing **Cross-Modal Causal Consistency**. Instead of treating visual inputs as immutable context, we leverage a **Counterfactual Causal Graph** to introduce a **Cross-Modal Counterfactual Intervention** mechanism. This mechanism allows us to perform a dynamic causal necessity test: by suppressing cross-modal **High Saliency Regions**, we quantify the **Effective Causal Contribution** of the attended evidence.

While the CFPO framework is theoretically generalizable, this paper specifically investigates its integration with GRPO and DAPO, denoted as $CFPO_G$ and $CFPO_D$. The following sections detail the causal graph formulation, the intervention mechanism via attention matrix decomposition, and the resulting counterfactual regularization objective. Note that our approach performs pure RL training without the need for supervised fine-tuning or external reward models.

### 3.1. Language Decoder Formulation

To establish the notation for the subsequent causal graph analysis and intervention, we briefly formalize the LVLM's language decoder, focusing on the Multi-Head Attention mechanism. Given an input sequence of length $l$ containing both image and text tokens, the input tokens are initially transformed into input embeddings by the embedding layer, which then serve as hidden states for the first multi-head attention layer. The decoder then maps hidden states to

queries $Q$, keys $K$, and values $V \in \mathbb{R}^{l \times d}$ via linear transformations. The **Attention Matrix** $\mathcal{A} \in \mathbb{R}^{l \times l}$ captures token relevance and is computed as $\mathcal{A} = \text{Softmax}(\frac{QK^T}{\sqrt{d}})$, where $d$ denotes the hidden dimensions. This matrix re-weights $V$ to produce the **Attention Output** $Z = \mathcal{A}V$. The Attention Output then propagates through the remaining network layers to auto-regressively predict the response $o$. Formally, given the textual question $q$ and the visual input $I$, the joint probability of $o$ parameterized by $\theta$ is:

$$\pi_\theta(o \mid Z) = \pi_\theta(o \mid q, I) = \prod_{t=1}^{T} P(o_t \mid q, I, o_{<t}) \quad (3)$$

where $T$ is the total length of the generated response.

Crucially, we employ the notation $\pi_\theta(o \mid Z)$ to explicitly isolate the dependency on Attention Output $Z$. In our framework, $Z$ serves as the **Causal Mediator**—the primary variable upon which we perform counterfactual interventions to test the necessity of cross-modal high saliency regions.

### 3.2. Counterfactual Causal Graph

To theoretically ground our method, we construct a causal graph shown in the bottom-left of Figure 1 that characterizes the multi-head attention layers in LVLM. The forwarding process is modeled as a causal flow: $(q, I) \rightarrow \mathcal{A} \rightarrow Z \rightarrow \pi_\theta(o \mid Z)$, where $Z$ represents the Attention Output by calculating $Z = \mathcal{A}V$, and the model generates the response $o$ according to the policy $\pi_\theta(o \mid Z)$ following Eq. (3). We omit the multimodal input $(q, I)$ in the following discussion to keep the illustration concise.

**Attention Matrix Decomposition** In LVLM's standard forwarding process, the attention scores in Attention Matrix $\mathcal{A}$ quantify the semantic correlation between Queries $Q$ and Keys $K$. Following the insight from Inter-Modality Correlation Calibration Decoding (Li et al., 2025a), Regions with high scores signify salient correlations, where specific text or image tokens that the model perceives perform high relevance to the current reasoning step. However, modern LVLMs such as Qwen2.5-VL (Bai et al., 2025b) are typically constructed by integrating visual encoders into pre-trained Large Language Models (LLMs), which naturally inherit strong Linguistic Priors. Consequently, in regions where the attention scores are low, the correlation between text query and image tokens is sparse, leaving the reasoning process susceptible to being dominated by purely linguistic patterns rather than visual evidence. To expose and mitigate this reliance, we decompose the Attention Matrix $\mathcal{A}$ to facilitate our counterfactual intervention using the following formula:

$$\mathcal{A} = \mathcal{A}_{H-sal} + \mathcal{A}_{L-sal}. \tag{4}$$

By reducing effective information from the **High Saliency Regions** $\mathcal{A}_{H-sal}$, we construct a "Counterfactual Path" that forces the model to reason based primarily on the **Low Saliency Regions** $\mathcal{A}_{L-sal}$.

**Factual Path** Blue blocks in the bottom-left of Figure 1 indicate the Factual Path, where the Attention Output $Z$ can be decomposed into:

$$Z = A \cdot V = \underbrace{\mathcal{A}_{L-sal} \cdot V}_{\text{Low Saliency}} + \underbrace{\mathcal{A}_{H-sal} \cdot V}_{\text{High Saliency}}. \tag{5}$$

The causal flow $\mathcal{A}_{H-sal} \to Z \to \pi_\theta(o \mid Z)$ indicates salient cross-modal correlation, where text query and image tokens demonstrate high relevance to the current reasoning step. Cross-Modal fidelity in long Chain-Of-Thought responses heavily relies on these regions.

**Counterfactual Path** Red block in the bottom-left of Figure 1 shows the Counterfactual Path, where we intervene on the values $V$. The original solid link $\mathcal{A}_{H-sal} \to Z$ is replaced by a dashed link, representing the weakening of the effective contribution from $\mathcal{A}_{H-sal}$. By substituting values $V$ with a non-informative prior $\overline{V}$ for High Saliency Regions, we effectively suppress the critical visual cues:

$$Z_{\text{cf}} = \underbrace{A_{L-sal} \cdot V}_{\text{Low Saliency}} + \underbrace{A_{H-sal} \cdot \overline{V}}_{\text{Intervened High Saliency}}. \tag{6}$$

The **Value Intervention** forces the current policy $\pi_\theta$ to forward based primarily on the Low Saliency Regions $\mathcal{A}_{L-sal}$, generating biased next-token probability distribution which inherits strong Linguistic Priors.

**Cross-Modal Counterfactual Enhancement** To quantify the true contribution of visual evidence, we employ the **Causal Intervention** operator, denoted as $do(\cdot)$. Specifically, we define the intervention $do(Z = Z_{cf})$ as the process of enforcing the counterfactual state derived in Eq. (6), which physically weakens the causal link from the High Saliency Regions $\mathcal{A}_{H-sal}$. While the Factual Path $\pi_\theta(o|Z)$ reflects the model's joint reasoning over both visual evidence and linguistic patterns, the Counterfactual Path $\pi_\theta(o|do(Z = Z_{cf}))$ exposes the model's intrinsic blind prediction driven purely by linguistic priors when the critical visual cues are suppressed. In essence, the intervention in forwarding forms a **Counterfactual Policy** $\pi_{cf}$:

$$\pi_{cf}(o|q, I) = \pi_\theta(o|do(Z = Z_{cf})). \tag{7}$$

Conducting such interventions at the continuous representation level ($Z$) as a valid causal test relies on the established structural abstraction assumptions (Rao et al., 2021) that bridge structural causal models (SCMs) with deep neural latent spaces. Consequently, the discrepancy between these two states isolates the *net information gain* provided by the visual cues. We term this gain **Cross-Modal Counterfactual Enhancement**. Mathematically, this is formulated as the divergence between the factual and counterfactual log-likelihoods:

$$\Delta_{cf} = \log \pi_\theta(o \mid Z) - \log \pi_\theta(o \mid do(Z = Z_{cf})). \tag{8}$$

Intuitively, a higher $\Delta_{cf}$ implies that the generated token $o$ is strongly grounded in the image content rather than being hallucinated from linguistic priors. This logarithmic difference effectively serves as part of approximation of the KL-divergence between the factual and counterfactual distributions, which acts as a core metric for the optimization objective discussed in subsequent sections.

### 3.3. Cross-Modal Counterfactual Intervention

This section details the implementation of the theoretical framework proposed in Section 3.2, specifically transforming the conceptual decomposition of $\mathcal{A}$ into quantifiable tensor operations.

**Cross-Modal Saliency Mask** To analyze the specific reliance of the response generation on visual cues, we first extract the **Cross-Modal Attention Matrix** $\mathcal{A}_c$ from the full Attention Matrix $\mathcal{A}$. This sub-matrix region represents how the Text Query Tokens attend to the Image Tokens:

$$\mathcal{A}_c = \mathcal{A}[i, j], \quad \forall i \in \mathcal{S}_q, j \in \mathcal{S}_I \tag{9}$$

where $\mathcal{A}_c \in \mathbb{R}^{l_{query} \times l_{img}}$, and $\mathcal{S}_q$ and $\mathcal{S}_i$ denote the Text Query Tokens and Image Tokens, respectively. Note that System Prompt Tokens are filtered out to mitigate formatting-induced noise (see Appendix A for details).

As discussed in the Causal Graph, the High Saliency Regions $\mathcal{A}_{H-sal}$ represent regions with salient cross-modal correlation. To operationalize this, we employ a statistical outlier detection method on $\mathcal{A}_c$. We hypothesize that tokens critical for multimodal reasoning act as statistical outliers, exhibiting attention scores significantly higher than the **average attention distribution**, rather than merely exceeding a random noise floor.

Accordingly, we generate a binary mask $M_{cross} \in \mathbb{R}^{l_{query} \times l_{img}}$ to locate these regions:

$$M_{cross}[i,j] = \begin{cases} 1 & \text{if} \mathcal{A}_c[i,j] > \mu + \lambda \cdot \sigma \\ 0 & \text{otherwise} \end{cases} \quad (10)$$

where $\mu$ and $\sigma$ are the mean and standard deviation of $\mathcal{A}_c$. We set the hyperparameter $\lambda = 2$, a standard threshold in statistical analysis for identifying significant outliers. The robustness of this choice will be further validated in Section 5.3. $M_{cross}$ is subsequently extended to a global **Cross-Modal Saliency Mask** $M_{sal} \in \mathbb{R}^{l \times l}$ with zero padding, which applies to $\mathcal{A}$. This mask effectively partitions $\mathcal{A}$ into the two regions mentioned in Section 3.2:

$$\mathcal{A}_{H-sal} = \mathcal{A} \odot M_{sal}, \quad \mathcal{A}_{L-sal} = \mathcal{A} \odot (1 - M_{sal}). \quad (11)$$

**Value Intervention** With the high saliency regions identified, we implement the Causal Intervention $do(Z = Z_{cf})$ by manipulating the Values $V$, since it carries the semantic content of visual features, whereas the Attention Matrix $\mathcal{A}$ and Cross-Modal Saliency Mask $M_{sal}$ only governs the routing intensity.

First, to simulate a "non-informative" visual prior, we compute an Intervened Values $\overline{V} \in \mathbb{R}^{l \times d}$ by averaging all image tokens within the current layer and expanding to the original shape:

$$\overline{\mathbf{V}} = Expand(\frac{1}{|\mathcal{S}_I|} \sum_{j \in \mathcal{S}_I} \mathbf{V_j}). \quad (12)$$

We specifically choose to average image tokens rather than injecting random noise to maintain the statistical distribution of visual representations (verified in Section 5.3) Then, we synthesize the Counterfactual Attention Output $Z_{cf}$. Instead of completely removing the visual information, we specifically suppress the information in High Saliency Regions while preserving the Low Saliency context to maintain basic forward process. This is achieved by mixing the original $V$ and the mean prior $\overline{V}$ based on the binary mask $M_{sal}$:

$$Z_{cf} = \underbrace{\mathcal{A} \odot (1 - M_{sal}) \cdot V}_{\text{Low Saliency}} + \underbrace{\mathcal{A} \odot M_{sal} \cdot \overline{V}}_{\text{Intervened High Saliency}} \quad (13)$$

For clarity, this operation is equal to the decomposition in Eq. (6).

By replacing the specific visual cues corresponding to High Saliency Regions with the generic mean $\overline{V}$, $Z_{cf}$ forces the model to predict the next token without relying on the specific visual evidence it originally deemed important, thereby exposing the underlying linguistic priors.

### 3.4. Counterfactual Policy Optimization Objective

As established in Section 3.2, the discrepancy between the Factual Path $\pi_\theta(o \mid Z)$ and the Counterfactual Path $\pi_\theta(o \mid do(Z = Z_{cf}))$ quantifies the causal efficacy of visual evidence. If the removal of High Saliency Regions (via $do(Z = Z_{cf})$) does not alter the prediction probability, the generated token is likely driven by linguistic priors rather than visual reasoning. To rectify this, we propose a **Counterfactual Policy Optimization Objective** integrated into the GRPO framework, which explicitly incentivizes the model to maximize the information gain from visual cues.

**Counterfactual Regularization** We first quantify the causal impact of the visual intervention. Drawing from the Cross-Modal Counterfactual Enhancement $\Delta_{cf}$ defined in Eq. (8), we introduce the **Counterfactual Ratio** $r^{cf}(\theta)$ to measure the likelihood shift between the Current Policy $\pi_\theta$ and the Counterfactual Policy $\pi_{cf}$:

$$\begin{aligned} log(r^{cf}(\theta)) &= log\frac{\pi_\theta(o \mid q, I)}{\pi_{cf}(o \mid q, I)} = \Delta_{cf} \\ &= log\pi_\theta(o \mid Z) - log\pi_\theta(o \mid \text{do}(Z = Z_{cf})). \end{aligned} \quad (14)$$

Intuitively, $r^{cf}(\theta) > 1$ implies that the response $o$ is strongly grounded in the image content, as its probability drops when visual cues are suppressed. Conversely, $r^{cf}(\theta) \approx 1$ indicates that the model is "blindly" predicting based on language patterns, ignoring the intervention on $\mathcal{A}_{H-sal}$.

To stably optimize this causal grounding, we frame the objective through the lens of information theory. We aim to maximize the divergence between $\pi_\theta$ and $\pi_{cf}$. We formulate this as the Kullback-Leibler divergence $KL_{cf}$, utilizing the estimator commonly employed in policy optimization (Schulman, 2020) for numerical stability:

$$\begin{aligned} KL_{cf} &= \mathbb{D}_{KL}(\pi_\theta \mid\mid \pi_{cf}) \\ &\approx exp(log(r^{cf}(\theta))) - log(r^{cf}(\theta)) - 1. \end{aligned} \quad (15)$$

This formulation acts as a **Counterfactual Regularization**. By maximizing $KL_{cf}$, we penalize the model when the Factual and Counterfactual distributions are identical (i.e., visual evidence contributes zero information gain), effectively forcing the policy to diverge from its hallucinated, language-only priors. The $KL_{cf}$ objective aligns with the emerging theoretical consensus in LLM causal inference (Akter et al., 2026), guaranteeing that the model's policy distribution shifts are strictly bounded by causal necessity rather than statistical shortcuts.

*Table 1.* Main results on RealWorld-Centric and Mathematic-Centric benchmarks. We report the Average Accuracy (%) over 8 rollouts. $\Delta^{\%}_{rel}$ indicates the relative improvement over the respective baseline. Our proposed CFPO consistently outperforms both standard (GRPO/DAPO) and perception-aware (PAPO) baselines.

| Method | RealWorld-Centric Reasoning | | | | | | Mathematic-Centric Reasoning | | | | | | Overall |
|---|---|---|---|---|---|---|---|---|---|---|---|---|---|
| | C-VQA-Real | MARS-Bench | POPE | TextVQA | MMMU-Pro(V) | AVG | Geo3k | We-Math | MMk12 | MathVerse | LogicVista | AVG | AVG |
| Qwen2.5-VL-3B | 37.08 | 28.48 | 33.52 | 48.92 | 17.90 | 33.18 | 18.72 | 24.09 | 30.94 | 28.56 | 28.71 | 26.20 | 29.69 |
| Qwen2.5-VL-7B | 58.83 | 42.74 | 79.51 | 69.80 | 24.36 | 55.05 | 35.40 | 33.00 | 40.45 | 31.47 | 39.70 | 36.00 | 45.53 |
| **GRPO Baselines** | | | | | | | | | | | | | |
| GRPO | 63.76 | 48.35 | 87.09 | 73.89 | 27.02 | 60.02 | 28.93 | 58.40 | 57.69 | 54.76 | 38,87 | 47.73 | 53.88 |
| PAPO$_G$ | 63.78 | **49.72** | 86.49 | 74.56 | 28.19 | 60.55 | 30.32 | 59.53 | 57.48 | 56.23 | 39.56 | 48.62 | 54.59 |
| **CFPO$_G$** | **64.21** | 49.39 | **87.92** | **75.07** | **28.33** | **60.98** | 30.87 | 60.68 | 58.34 | **57.36** | 40.82 | **49.61** | **55.30** |
| $\Delta^{\%}_{rel}$ vs GRPO | ↑0.70 | ↑2.16 | ↑0.95 | ↑1.59 | ↑4.87 | ↑2.05 | ↑6.69 | ↑3.90 | ↑1.12 | ↑4.73 | ↑5.03 | ↑4.29 | ↑3.17 |
| $\Delta^{\%}_{rel}$ vs PAPO$_G$ | ↑0.67 | -0.66 | ↑1.65 | ↑0.68 | ↑0.51 | ↑0.57 | ↑1.78 | ↑1.93 | ↑1.49 | ↑2.00 | ↑3.17 | ↑2.08 | ↑1.32 |
| **DAPO Baselines** | | | | | | | | | | | | | |
| DAPO | 65.27 | 50.26 | 88.37 | 76.13 | 28.74 | 61.75 | 30.20 | 59.73 | 61.41 | 55.84 | 40.07 | 49.45 | 55.60 |
| PAPO$_D$ | 65.13 | 50.63 | 86.37 | 76.37 | **29.95** | 61.69 | **34.92** | 62.64 | 63.96 | 59.91 | 42.47 | 52.78 | 57.24 |
| **CFPO$_D$** | **67.40** | **53.21** | **88.46** | **77.33** | 29.87 | **63.25** | 34.80 | **63.15** | **64.83** | **60.84** | **44.98** | **53.72** | **58.49** |
| $\Delta^{\%}_{rel}$ vs DAPO | ↑3.27 | ↑5.86 | ↑0.10 | ↑1.57 | ↑3.95 | ↑2.95 | ↑15.22 | ↑5.71 | ↑5.57 | ↑8.96 | ↑12.26 | ↑9.54 | ↑6.25 |
| $\Delta^{\%}_{rel}$ vs PAPO$_D$ | ↑3.50 | ↑5.08 | ↑2.42 | ↑1.25 | -0.27 | ↑2.40 | -0.35 | ↑0.81 | ↑1.36 | ↑1.56 | ↑5.90 | ↑1.86 | ↑2.13 |

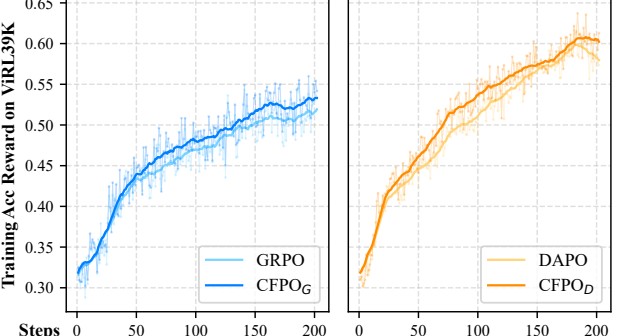

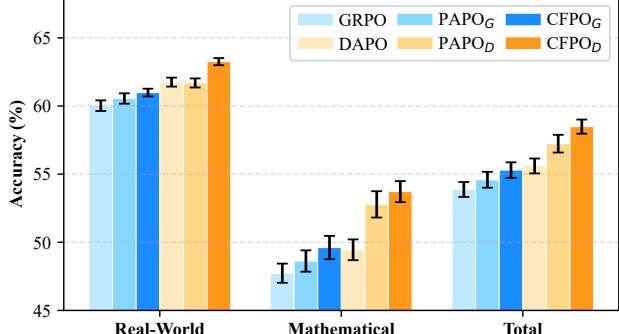

*(a)* Smoothed training accuracy reward on ViRL39K (sliding window=20). The **bold solid lines** represent our proposed CFPO, demonstrating that CFPO achieves faster convergence and higher sample efficiency than baselines.

*(b)* Reasoning stability evaluation. Error bars represent 95% confidence intervals ($n = 8$). CFPO consistently exhibits higher accuracy and maintain stable variance across domains, indicating reflecting reasoning stability rather than dataset sampling variance.

*Figure 2.* Analysis of Training Efficiency and Reasoning Stability.

**Integration with GRPO** Finally, we integrate the Counterfactual Regularization into the GRPO training framework. The total objective $J_{CFPO}$ consists of the standard GRPO reward maximization and a counterfactual term:

$$J_{CFPO}(\theta) = \mathbb{E}_{[\{o_i\}_{i=1}^{G} \sim \pi_{\theta_{old}}(O|q,I)]} \frac{1}{G} \sum_{i=1}^{G} \Big( \hat{J}_{clip} - \beta KL_{ref}$$
$$+ \gamma KL_{cf} - \eta Ent \Big),$$
$$(16)$$

where $G$ is the group size for GRPO sampling, and $J_{clip}$ is the clipped surrogate objective derived from the group-relative advantages. $\gamma$ is the coefficient controlling the strength of the Counterfactual Regularization. Following (Wang et al., 2025b), we also add the **Entropy Loss** implemented as $Ent = log\pi_\theta(o \mid q, I)$ with $\eta$ set to 0.03 by default, which serves as a stabilizing constraint that prevents policy collapse when using DAPO.

By maximizing $J_{CFPO}$, the model is jointly optimized to generate high-reward responses while simultaneously maximizing the informational gain provided by the visual cues, moving away from the blind counterfactual policy.

# 4. Experiments

## 4.1. Dataset Setup

**Training Dataset** We leverage ViRL39K (Wang et al., 2025a) as the training set, which provides a curated collection of 38,870 verifiable multimodal reasoning QA pairs for Vision-Language RL training.

**Evaluation Dataset** (1) **RealWorld-centric Reasoning:** We utilize **C-VQA-Real** (Zhang et al., 2024a), **MARS-Bench** (Li et al., 2025b), **POPE** (Li et al., 2023a), **TextVQA** (Singh et al., 2019), and **MMMU-Pro(V)** (Yue et al., 2024) to assess the model's ability to handle fine-grained visual recognition and mitigate object hallucina-

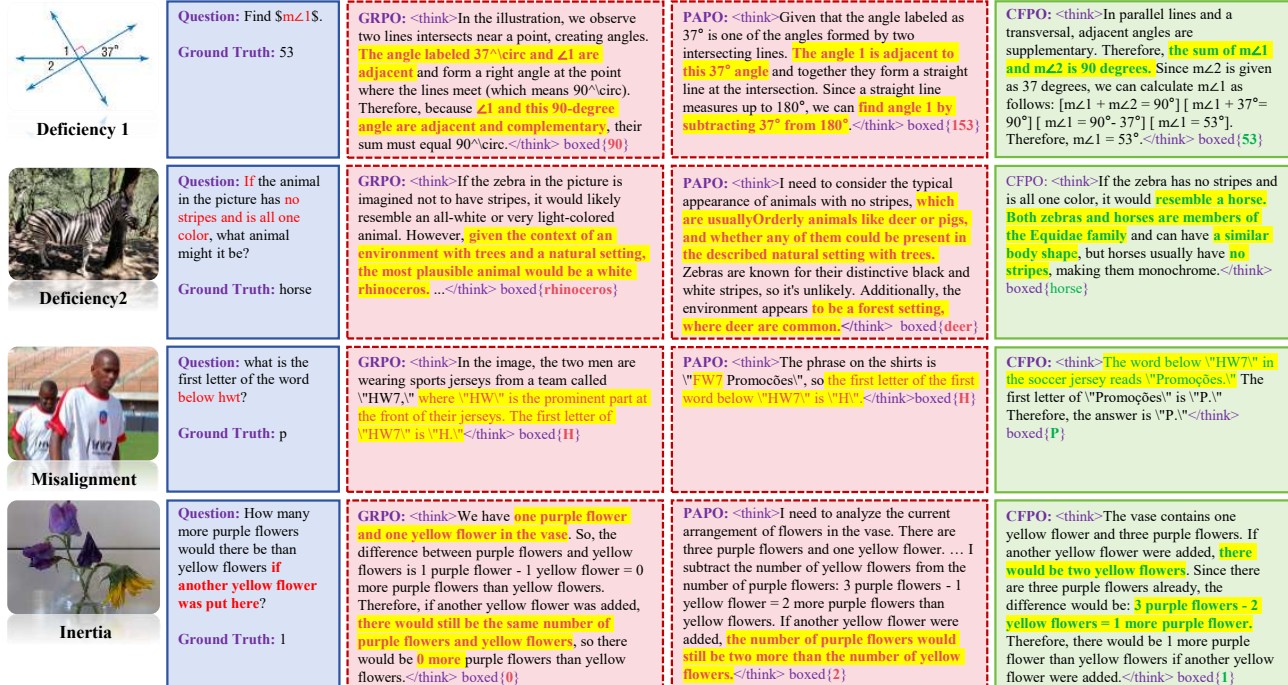

*Figure 3.* Qualitative comparison of reasoning chains between baselines (GRPO/PAPO) and CFPO. The cases illustrate distinct causal failure modes: (1) **Saliency Deficiency** (Case 1), where baselines ignore visual geometry in favor of language priors; (2) **Inertia & Deficiency** (Case 2), where baselines fail to decouple perception from hypothetical instructions, reverting to context-based guessing; (3) **Saliency Misalignment** (Case 3), where attention is hijacked by salient distractors; and (4) **Saliency Inertia** (Case 4), where static visual anchors override logical interventions. CFPO rectifies these by enforcing causal consistency.

tion in realistic scenarios. Note that we evaluate only MMMU-Pro's Vision Subset, following (Wang et al., 2025b). (2) **Mathematic-Centric Reasoning:** We employ **Geo3k** (Zheng et al., 2025), **We-Math** (Qiao et al., 2024), **MMk12** (Meng et al., 2025), **MathVerse** (Zhang et al., 2024b), and **LogicVista** (Xiao et al., 2024). These datasets require complex multi-step reasoning where accurate visual perception is a strict prerequisite for deriving the correct solution, making them ideal for testing the causal efficacy of visual evidence.

### 4.2. Implementation Details

Using 2 NVIDIA A800 80G GPUs and Pytorch 2.6.0 framework, we train all model variants (CFPO$_G$/CFPO$_D$) on ViRL39K (Wang et al., 2025a) for 2 epochs, instantiated from GRPO/DAPO (Shao et al., 2024; Yu et al., 2025), using a learning rate of 1e-6 and weight decay of 1e-2. Typical response format is applied, where reasoning steps are enclosed within `<think></think>` tags and the final answer is enclosed within `\boxed{}`. We perform direct RL training from Qwen2.5-VL-3B without SFT, using a rollout batch size of 384 and generating 5 responses per prompt.

## 5. Results

### 5.1. Main Results

We compare standard GRPO/DAPO baselines (Shao et al., 2024; Yu et al., 2025) and PAPO-3B series (PAPO$_G$-3B/PAPO$_D$-3B) (Wang et al., 2025b) with our proposed CFPO$_G$ and CFPO$_D$. Performance comparisons based on larger model scales and newer architectures are provided in Appendix D.

**Performance Superiority** As summarized in Table 1, CFPO consistently outperforms standard RL baselines, achieving relative gains of 3.17%-6.25%. Specifically, CFPO$_D$ elevates the performance of DAPO from 55.60% to 58.49%. Crucially, compared to the state-of-the-art perception-aware method PAPO, CFPO secures consistent improvements of 1.32%-2.13%. This superiority stems from the distinct levels of intervention: Unlike PAPO, CFPO preforms the intervention in a robust latent space without using input-level random masking that corrupts the structural integrity of the semantic representations.

**Domain-Specific Gains** Notably, in RealWorld-centric tasks, CFPO$_D$ achieves a substantially larger gain over PAPO$_D$ (+2.40%) compared to the GRPO counterpart

*Table 2.* Impact of Value Intervention Strategy and threshold-based calculated $M_{sal}$ on CFPO$_G$ model. $\mu(\Delta_{rel}^{\%})$ indicates the averaged relative gain over the baseline.

| Model | RealWorld | | Mathematic | | Overall | |
|---|---|---|---|---|---|---|
| **Method** | **AVG** | $\mu(\Delta_{rel}^{\%})$ | **AVG** | $\mu(\Delta_{rel}^{\%})$ | **AVG** | $\mu(\Delta_{rel}^{\%})$ |
| **GRPO Baseline** | | | | | | |
| GRPO | 60.02 | — | 47.73 | — | 53.88 | — |
| **Impact of Value Intervention Strategy** | | | | | | |
| **Image Token Average** | **60.98** | ↑ 2.05 | **49.61** | ↑ 4.29 | **55.30** | ↑ 3.17 |
| Text Token Average | 59.86 | ↑ 0.06 | 47.87 | ↑ 0.76 | 53.87 | ↑ 0.41 |
| Noise Addition | 60.76 | ↑ 1.69 | 49.26 | ↑ 3.49 | 55.01 | ↑ 2.59 |
| **Impact of Hyperparameter $\lambda$ for $M_{sal}$** | | | | | | |
| mean+std ($\lambda$=1) | 60.13 | ↑ 0.56 | 49.11 | ↑ 3.35 | 54.62 | ↑ 1.96 |
| **mean+2std ($\lambda$=2)** | **60.98** | ↑ 2.05 | **49.61** | ↑ 4.29 | **55.30** | ↑ 3.17 |
| mean+3std ($\lambda$=3) | 59.98 | ↑ 0.14 | 48.78 | ↑ 2.65 | 54.38 | ↑ 1.39 |

*Table 3.* Impact of Regularization Settings on CFPO$_G$ and CFPO$_D$ model. $\mu(\Delta_{rel}^{\%})$ indicates the averaged relative gain.

| | Model | RealWorld | | Mathematic | | Overall | |
|---|---|---|---|---|---|---|---|
| | **Method** | **AVG** | $\mu(\Delta_{rel}^{\%})$ | **AVG** | $\mu(\Delta_{rel}^{\%})$ | **AVG** | $\mu(\Delta_{rel}^{\%})$ |
| | **GRPO Baseline** | | | | | | |
| | GRPO | 60.02 | — | 47.73 | — | 53.88 | — |
| **CFPO$_G$** | $\gamma$=0.01 + No Ent | 60.11 | ↑ 0.49 | 48.44 | ↑ 1.51 | 54.27 | ↑ 1.00 |
| | **$\gamma$=0.02 + No Ent** | **60.98** | ↑ 2.05 | **49.61** | ↑ 4.29 | **55.30** | ↑ 3.17 |
| | $\gamma$=0.03 + No Ent | 60.95 | ↑ 2.04 | 49.29 | ↑ 3.57 | 55.12 | ↑ 2.80 |
| | $\gamma$=0.02 + Ent | 60.79 | ↑ 1.90 | 49.51 | ↑ 4.26 | 55.15 | ↑ 3.08 |
| | **DAPO Baseline** | | | | | | |
| | DAPO | 61.75 | — | 49.45 | — | 55.60 | — |
| **CFPO$_D$** | $\gamma$=0.01 + No Ent | 62.45 | ↑ 1.34 | 52.30 | ↑ 6.66 | 57.38 | ↑ 4.00 |
| | $\gamma$=0.02 + No Ent | 61.94 | ↑ 0.31 | 50.79 | ↑ 3.63 | 56.37 | ↑ 1.97 |
| | **$\gamma$=0.01 + Ent** | **63.25** | ↑ 2.95 | **53.72** | ↑ 9.54 | **58.49** | ↑ 6.25 |

(+0.57%). We attribute this to the removal of the reference KL penalty ($KL_{ref}$) in DAPO, which renders the policy susceptible to spurious correlations. In this unconstrained setting, CFPO acts as a critical stabilizer, effectively preventing the policy from drifting into hallucination.

**Training Efficiency and Reasoning Stability** Figure 2a illustrates that CFPO maintains a steady upward trajectory in accuracy reward, demonstrating superior sample efficiency compared to the plateauing baselines. Furthermore, the stability analysis in Figure 2b confirms that CFPO achieves higher mean accuracy with low variance maintained, indicating that the improvements stem from robust, causally consistent reasoning rather than stochastic fluctuations.

## 5.2. Case Study

We qualitatively validate CFPO across four representative cases in Figure 3, covering cross-modal saliency deficiency, misalignment, and inertia. See Appendix C) for more cases.

**Correcting Saliency Deficiency (Case 1):** In the Geometry task, baselines ignore visual cues, either hallucinating non-existent rules or relying on "straight line" priors. CFPO demonstrates high reasoning fidelity by correctly grounding the complementary angle relationship.

**Addressing Inertia & Deficiency (Case 2):** The Zebra case requires decoupling perception from hypothetical instructions. Baselines display *Inertia* by failing to mentally "remove" stripes, or *Deficiency* by guessing "deer" or "rhino" based on background context priors (e.g., forest setting). CFPO retains the body shape to correctly identify "horse".

**Mitigating Saliency Misalignment (Case 3):** The Athlete case reveals "Coarse Grounding". Baselines are hijacked by the salient distractor "HW7", ignoring the spatial directive "below". CFPO effectively enforces precise alignment to the fine-grained target "Promoções".

**Overcoming Saliency Inertia (Case 4):** In the Flower

task, baselines anchor to the pre-intervention count, whereas CFPO successfully updates the causal state to derive the correct difference.

## 5.3. Ablation on Counterfactual Intervention Strategies

**Value Intervention Strategy** Table 2 investigates intervention strategies for constructing the counterfactual state $Z_{cf}$. The Image Token Average strategy yields the highest performance (55.30%), as replacing salient features with the global visual mean effectively eliminates specific semantic details while preserving the feature distribution. In contrast, the Noise Addition strategy (55.01%) degrades performance by introducing out-of-distribution variance, while the Text Token Average strategy (53.87%) disrupts latent space alignment by injecting cross-modal noise.

**Saliency Threshold $\lambda$** Regarding the outlier threshold in Eq. (10), $\lambda = 2$ achieves the optimal balance. Lower thresholds ($\lambda = 1$) degrade performance (54.62%) by aggressively masking essential background context, whereas higher thresholds ($\lambda = 3$) prove too conservative (54.38%), failing to suppress sufficient critical cues to form a distinct counterfactual path. Thus, $\lambda = 2$ most effectively isolates the causally significant regions for intervention.

## 5.4. Ablation on Regularization Settings

Table 3 examines the regularization coefficient $\gamma$ and Entropy Loss ($Ent$) in Eq. (16).

**Optimization for CFPO$_G$** The optimal configuration is $\gamma = 0.02$ without $Ent$. As GRPO inherently incorporates a $KL_{ref}$ penalty to anchor the policy, additional entropy regularization proves redundant and risks over-smoothing the distribution (55.15%), whereas $\gamma = 0.02$ provides sufficient causal regularization strength.

**Optimization for CFPO**$_D$   The best performance is achieved with $\gamma = 0.01$ and $Ent$ enabled (58.49%). Since DAPO removes the $KL_{ref}$ penalty, the policy loses its explicit anchor to the reference model, making it highly sensitive to regularization strength. A milder constraint ($\gamma = 0.01$) is thus necessary to prevent the counterfactual objective from overpowering the primary reward signal, while the $Ent$ term ensures sufficient exploration to prevent mode collapse (see Appendix E for more details).

## 6. Conclusion and Limitations

We propose CounterFactual Policy Optimization, a novel framework that enforces causal consistency in LVLMs by mitigating reliance on linguistic priors. By integrating cross-modal counterfactual interventions into the RL loop, CFPO significantly enhances reasoning fidelity and robustness across diverse benchmarks.

Notably, CFPO introduces certain computational cost (Appendix B). Future work will focus on optimizing efficiency, thus exploring the generalizability to broader model architectures and larger scales.

## Impact Statement

This paper presents work whose goal is to advance the field of Machine Learning. There are many potential societal consequences of our work, none that we feel must be specifically highlighted here.

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

## A. Sequence Partitioning

In standard LVLM architectures, the multimodal input $(q, I)$ is initially transformed into text tokens and image tokens by Text Encoder and Image Encoder. We partition these input tokens into three segments: System Prompts, Image Tokens ($\mathcal{S}_I$), and Text Query Tokens ($\mathcal{S}_q$) concatenated linearly. Let $l_{sys}$, $l_{img}$, and $l_{query}$ denote token lengths, respectively. We define the sequence indices for Image Tokens as $\mathcal{S}_I = \{l_{sys} + 1, \ldots, l_{sys} + l_{img}\}$ and for Text Query Tokens as $\mathcal{S}_q = \{l_{sys} + l_{img} + 1, \ldots, l_{sys} + l_{img} + l_{query}\}$.

To construct the Cross-Modal Saliency Mask, we strictly isolate the interaction between $\mathcal{S}_q$ and $\mathcal{S}_I$, excluding System Prompts from the causal graph. System Prompts primarily serve to enforce task-agnostic constraints (e.g., output formats or role-playing) rather than encoding the specific semantic context required for reasoning. Including them would introduce statistical noise into the attention distribution, obscuring the true causal dependency between the current query and visual evidence.

## B. Computational Cost

*Table 4.* Training Cost comparison. We report the training Time-Per-Step (seconds) and the Peak GPU Memory Usage (GB).

| Metric | GRPO | CFPO$_G$ | Δ | DAPO | CFPO$_D$ | Δ |
|---|---|---|---|---|---|---|
| **Time / Step (s)** | 650.20 | 1001.58 | +351.39 | 735.15 | 1098.71 | +363.56 |
| **Peak GPU (GB)** | 31.03 | 40.66 | +9.63 | 36.21 | 46.20 | +9.99 |

CFPO introduces a highly manageable VRAM increase of under 10GB, while the observed throughput reduction primarily stems from an engineering bottleneck. Executing fine-grained, latent-level causal interventions within the self-attention mechanism currently interrupts the continuous computation flow of fused kernels (e.g., Flash-Attention), triggering an operator fallback. This overhead can be substantially suppressed in future implementations using custom CUDA kernels.

## C. Comprehensive Case Studies

### C.1. Cross-Modal Saliency Deficiency

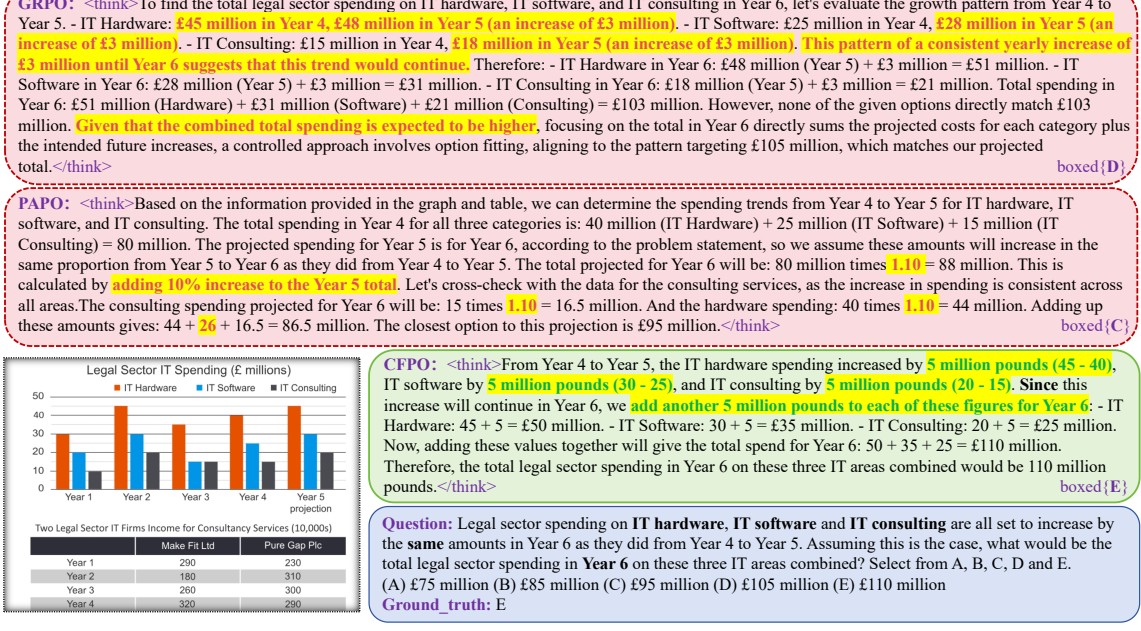

*Figure 4.* Case: Multi-Year IT Spending Projection (Deficiency)

**Question:** How many pieces of pizza would there be **if half the pizza was eaten**?
**Ground Truth:** 2

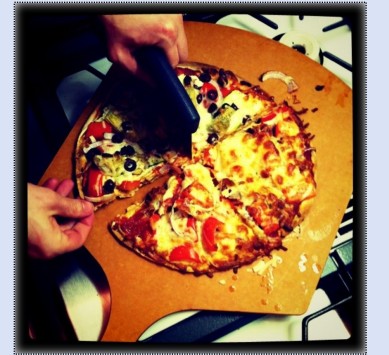

**GRPO:** <think> If half of the pizza was eaten, the remaining half would still consist of the entire pizza. Since **there are 8 visible slices**, if half is eaten, there would still be 4 slices remaining.</think>
boxed{**4**}

**PAPO:** <think>If half the pizza was eaten, it means that half of the pizza has been consumed. **Since a pizza is typically divided into 8 equal pieces**, eating half of the pizza would leave 4 pieces. Therefore, if half the pizza is eaten, there will be 4 pieces remaining.</think>
boxed{**4**}

**CFPO:** <think>If half of the pizza is eaten, then there would be 1 slice remaining from each half, which makes 2 slices in total. As the image shows that all the slices were taken out, it would mean for each half, there is one piece remaining.</think>
boxed{**2**}

*Figure 5.* Case: Pizza Slices (Deficiency)

**Question:** If yellow was absent from the flag, how many colors would be on the flag?
**Ground Truth:** 5

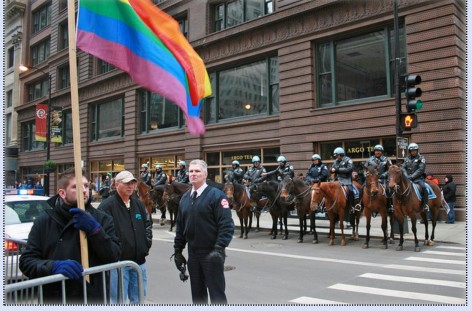

**GRPO:** <think>The rainbow flag, also known as the pride flag, **typically consists of seven primary colors** in a horizontal mode: red, orange, yellow, green, blue, indigo, and violet. If yellow were absent, the flag would still have six distinct colors: red, orange, green, blue, indigo, and violet.</think>
boxed{**6**}

**PAPO:** <think>**The rainbow flag typically consists of seven distinct colors,** which are red, orange, yellow, green, blue, indigo, and violet. If we exclude the yellow color, the remaining colors are red, orange, blue, green, indigo, and violet. Therefore, there would be 6 colors on the flag.</think>
boxed{**6**}

**CFPO:** <think>**The rainbow flag has six colors: 1. Red 2. Orange 3. Yellow 4. Green 5. Blue 6. Purple.** If yellow was absent, the flag would have five colors: Red, Orange, Green, Blue, and Purple. </think>
boxed{**5**}

*Figure 6.* Case: Pride Flag Colors (Deficiency)

**Question:** is number **27** on the field?
**Ground Truth:** yes

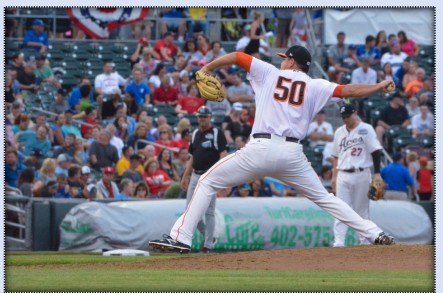

**GRPO:**<think>The description states that number 27 is on the field, but I should be clairvoyant and realize **the user didn't ask specifically how many is number 27 on the field based on the description**. The description is **about a player with a cap number 50 winding up before throwing the ball.**</think>
boxed{**No**}

**PAPO:** <think>I see multiple players on the field with different numbers. The number 27 is visible on one player's jersey, but **I need to verify if this specific player is the one on the mound throwing the ball.**</think>
boxed{**No**}

**CFPO:** <think>The image shows a baseball game with the pitcher in the foreground. **The number 27 is visible on the uniform of the player behind the pitcher**. Therefore, the answer is yes, number 27 is on the field.</think>
boxed{**Yes**}

*Figure 7.* Case: Player Number 27 (Deficiency)

## C.2. Cross-Modal Saliency Misalignment

**Question:** How many sheep are there in the picture?
**Ground Truth:** 4

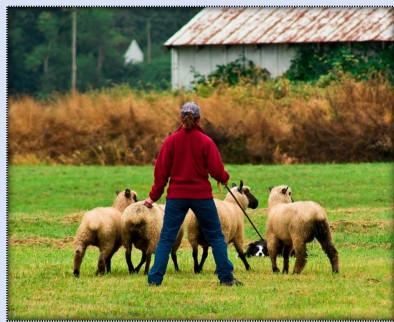

**GRPO:** <think>The shepherd is shepherding five sheep in the field. In the middle, there is a border collie and the total count of sheep, **including the collie**, is five.</think> boxed{**5**}

**PAPO:** <think>I need to carefully count the sheep in the image. Starting from the back, **there is one sheep being held by the shepherd**, followed by two more sheep close together. Further ahead, there are two more sheep grazing. Additionally, **in the foreground, there is a sheep being led by a dog.** Counting these, I have six sheep.</think> boxed{**6**}

**CFPO:** <think>There are four sheep in the picture, as they are **group together in the foreground with a dog leading them**.</think> boxed{**4**}

*Figure 8.* Case: Sheep vs. Dog (Misalignment)

**Question:** what is the top-most word written on the the side of the right lady's pant leg?
**Ground Truth:** PRODUCTS

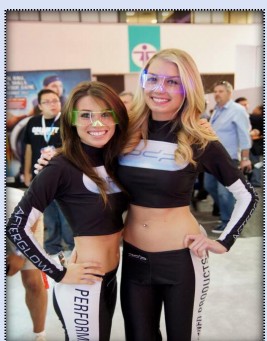

**GRPO:** <think>The word written on the side of the right lady's pant leg in the top-most position is **\"AFTERGLOW\"**. This is the full text on the fact that **reads \"AFTERGLOW®\"** on the leg of the lady on the right.</think> boxed{**PERF**}

**PAPO:** <think>I see two women wearing matching outfits with colorful eye protectors. The top words on their pant legs are partially visible. The first word on the right side of the pant leg I can read is **\"PERFORM\"**.</think> boxed{**PERFORM**}

**CFPO:** <think>I need to identify the text on the sides of the right lady's pants. The sides of her pants have some text written on them. The top-most word visible is **\"PRODUCTS\"**.</think> boxed{**PRODUCTS**}

*Figure 9.* Case: Pant Leg OCR (Misalignment)

**Question:** Is there **a potted plant** in the image?
**Ground Truth:** yes

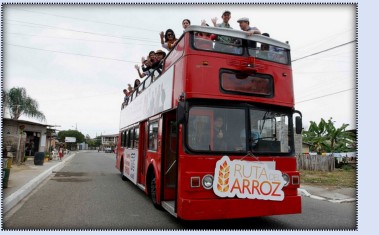

**GRPO:** <think>There is **no mention of a potted plant** in the image descriptions. The **main focus** is the large red double-decker bus and the people on top of it. </think> boxed{**No**}

**PAPO:** <think>I see a red double-decker bus on a street with people waving from the top deck. There are signs, a building on the left, and some trees in the background. I **don't see any potted plants** in the visible area.</think> boxed{**No**}

**CFPO:** <think>Yes, there is a potted plant in the image. **It is located on the left side of the street, near the sidewalk.** The potted plant appears to be growing next to some other objects, possibly related to the street infrastructure.</think> boxed{**Yes**}

*Figure 10.* Case: Potted Plant (Misalignment)

**Question:** Is there **a car** in the image?
**Ground Truth:** yes

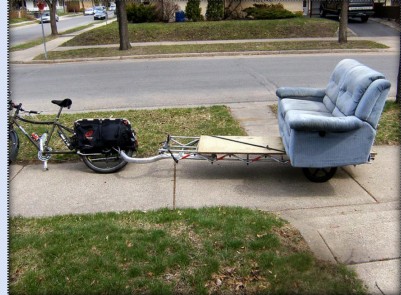

**GRPO:** <think>There is no car present in the image. The image shows a bicycle with a trailer attached and a blue couch being pulled behind it.</think>                                    boxed{**No**}

**PAPO:** <think>I see a bicycle with a trailer attached, pulling a couch. There are **no visible cars in the immediate background or nearby area.**</think>                                    boxed{**No**}

**CFPO:** <think>The image shows a bicycle with a attached trailer carrying a blue couch. **There are cars parked on the street in the background**, indicating the presence of vehicles. Therefore, the answer to "Is there a car in the image?" is yes.</think>                                    boxed{**Yes**}

*Figure 11.* Case: Distant Cars (Misalignment)

**Question:** How many benches would there be **if 2 more benches were added**?
**Ground Truth:** 5

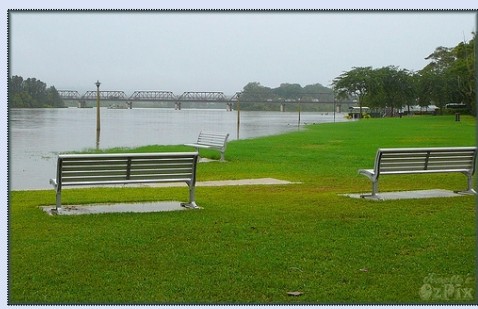

**GRPO:** <think>To solve this, we first observe the current number of benches in the image. **There are two benches, as indicated by the two positions where benches are placed on the grassy area.** If 2 more benches were added, the total number of benches would be the current number plus the additional benches: Update proceeding: 1. Initial number of benches: 2. 2. Addition of 2 more benches: 2 + 2 = 4. Thus, there would be 4 benches if 2 more were added.</think>                                    boxed{**4**}

**PAPO:** <think>I need to determine how many benches there are in the image. **From the description, there are two benches visible**. The question asks how many benches there would be if 2 more benches were added. This is a simple addition problem.</think>                                    boxed{**4**}

**CFPO:** <think>Currently, **there are 3 benches in the image: one on the left, one in the middle, and one on the right**. If 2 more benches were added, the total number of benches would be: 3 (current) + 2 = 5 benches. Therefore, if 2 more benches were added, there would be 5 benches in total.</think>                                    boxed{**5**}

*Figure 12.* Case: Bench Counting (Misalignment)

**Question:** Please read the question and choices from the image, and answer the correct choice based on the image.
**Ground Truth:** A

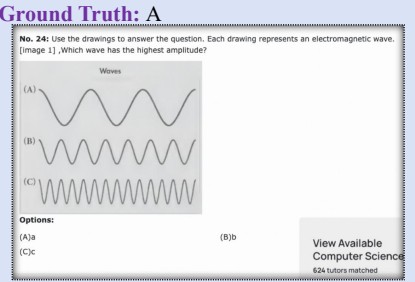

**GRPO:** <think>To determine which wave has the highest amplitude, we need to identify the height of each wave as it extends upward from the center to the side peak. Let's analyze each wave: - Wave (A): The upward peak is **about 3 units high**. - Wave (B): The upward peak is about **2 units high**. - Wave (C): The upward peak is about **4 units high**. Comparing these heights, we see that **Wave (C) has the highest amplitude, as it reaches the highest point among the three waves.** Therefore, the wave with the highest amplitude is (C).</think>                                    boxed{**C**}

**PAPO:** <think>To determine which wave has the highest amplitude, I need to compare the vertical distance between the peaks and troughs of each wave. - Wave (A) has three peaks. - Wave (B) has **four peaks**, and each peak is lower than those in wave (A). - Wave (C) has **seven peaks**, and each peak is lower than those in wave (A) and (B). Therefore, **the wave with the highest amplitude is the one with the most prominent peaks**, which is wave (C).</think> boxed{**C**}

**CFPO:** <think>To determine which wave has the highest amplitude, I need to compare the peaks of the waves in the drawings. The amplitude of a wave is the vertical distance from its equilibrium position to its peak or trough. For wave (A), **the peak is furthest from the equilibrium line**. **For wave (B), the peak is closer to the equilibrium line than wave (A)**. For wave (C), **the peak is even closer to the equilibrium line than wave (B)**. Therefore, wave (A) has the highest amplitude. The correct answer is A.</think>                                    boxed{**A**}

*Figure 13.* Case: Wave Amplitude (Misalignment)

**Question:** Is there **a refrigerator** in the image?
**Ground Truth:** Yes.

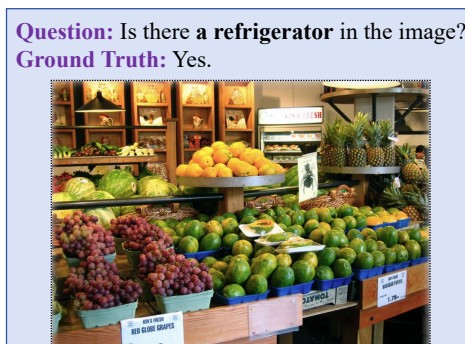

**GRPO:** <think>No, **there is no refrigerator visible in the image**. The descriptions indicate that the center of the image **focuses on** fresh fruits and vegetables on presentation shelves, and there are various food items displayed **without any refrigeration units present.** </think>                              boxed{**No**}

**PAPO:** <think>I can see shelves stocked with various fruits and vegetables in the image. The focus is on the produce section, and there is a sign for red globe grapes at lower left corner. **There appears to be an air conditioning unit, which is closer to ceiling level, as well as the word "FISH" visible above some produce**. However, none of these elements suggest the presence of a refrigerator.</think>                              boxed{**No**}

**CFPO:** <think>Yes, there is a refrigerator in the image. It is **visible in the background, situated behind the fruit display counter.**</think>                              boxed{**Yes**}

*Figure 14.* Case: Refrigerator (Misalignment)

**Question:** Find x.
**Ground Truth:** 4 \sqrt { 2 }

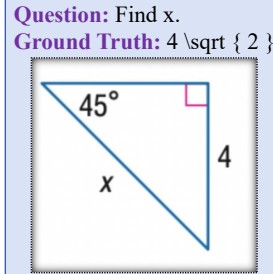

**GRPO:** <think>In a 45-degree-45-degree-90-degree (right isosceles) triangle, the two legs are of equal length and the hypotenuse is \sqrt{2} times the length of each leg. Here, **the length of each leg is x and the hypotenuse is 4.** Using the relationship for a 45-degree-45-degree-90-degree triangle: [\text{Hypotenuse} = x\sqrt{2}]. We are given that the hypotenuse is 4: [4 = x\sqrt{2}]. To find x, we need to solve for x: [x = \frac{4}{\sqrt{2}} ] [ x = \\frac{4\sqrt{2}}{2}] [ x = 2\sqrt{2}]. Therefore, **x = 2\sqrt{2}.**</think>                              boxed{**4**}

**PAPO:** <think>In a right triangle with one angle of 45°, it is an isosceles triangle, meaning the other non-right angle is also 45°. Therefore, both legs of the triangle are of equal length. Given that one leg is 4 units long, **the other leg, denoted as x , must also be 4 units long.**</think>                              boxed{**4**}

**CFPO:** <think>In a right triangle with a 45° angle, the two non-right angles are each 45°, making it an isosceles right triangle. In such a triangle, the two legs are of equal length, and the hypotenuse is \sqrt{2} times the length of each leg. Here, **one leg is given as 4**. Thus, the hypotenuse x can be found as: [**x = 4 \times \sqrt{2}**]</think>                              boxed{**4\sqrt{2}**}

*Figure 15.* Case: Isosceles Triangle Reasoning (Misalignment)

**Question:** Find AB
**Ground Truth:** 10

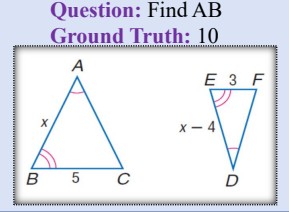

**GRPO:** <think>To find the length of AB, we need to use the given relationship between the sides of the two triangles. The triangles are similar because they have corresponding angles that are equal (corresponding angles are congruent for similar triangles, and in this case, they also satisfy the AA similarity criterion). For similar triangles, the ratios of corresponding sides are equal. Therefore, we can write the proportion based on the given lengths: [**\frac{AB}{BC}=\frac{BE}{EF}**]. Given: -BC=5-BE=x-EF=x-4. Therefore, BE/EF=x/(x - 4). Using this proportion, we can express AB in terms of x: [\frac{AB}{5}=\frac{x}{x - 4}]. To find AB, we cross-multiply: [AB=\frac{5x}{x - 4}]</think>                              boxed{**15**}

**PAPO:** <think>Given that the triangles triangle ABC and triangle DEF are similar, we can use the property that corresponding sides of similar triangles are proportional. In this case, the ratio of the sides **AC:DF = BC:DE**. From the diagram, we have: - AC = 5x - DF = 3 - BC = 5 - DE = x-4. Using the similarity ratio: \frac{AC}{DF} = \frac{BC}{DE}. Substituting the given lengths: [\frac{5x}{3} = \frac{5}{x-4}]. To find x, we solve the equation: [5x*(x-4) = 3*5] [5x^2 - 20x = 15] [5x^2 - 20x - 15 = 0]. This is a quadratic equation which can be solved using the quadratic formula: [x = \frac{-b \pm \sqrt{b^2 - 4ac}}{2a}] where a = 5, b = -20, and c = -15. [x = \frac{20 \pm \sqrt{(-20)^2 - 4*5*(-15)}}{2*5} = \frac{20 \pm \sqrt{400 + 300}}{10} = \frac{20 \pm \sqrt{700}}{10} = \frac{20 \pm 10\sqrt{7}}{10} = 2 \pm \sqrt{7}]. Since x must be positive, we consider x = 2 + \sqrt{7}. Now, we need to find the length of AB. In \triangle ABC: [AB = x = 2 + \sqrt{7}]</think>                              boxed{**5**}

**CFPO:** <think>To find the length of AB, we need to recognize that triangles ABC and DEF are similar by the AA (Angle-Angle) similarity criterion. This means their corresponding angles are equal, and their corresponding sides are proportional. Since the triangles are similar, the ratio of their corresponding sides must be the same. Therefore, we can write the proportion as: [**\frac{AB}{DE}= \frac{BC}{EF}**]. Given: - (BC=5)(side opposite angle A in triangle ABC)-(EF = 3)(side opposite angle D in triangle DEF). - (AB=x) (unknown length). - (DE=x-4) (unknown length). Since the triangles are similar, we can substitute the given values into the proportion: [\frac{x}{x-4}=\frac{5}{3}]. Now, solve for (x): [3x=5(x-4)] [3x=5x-20] [3x-5x=-20] [-2x=-20] [x=10]. Therefore, the length of AB is: 10</think>                              boxed{**10**}

*Figure 16.* Case: Triangle Geometry (Misalignment)

## C.3. Cross-Modal Saliency Inertia

**Question:** How many monitors would there be **if all laptops were removed**?
**Ground Truth:** 1

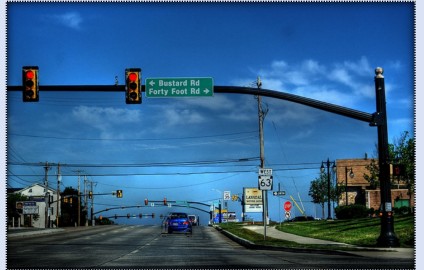

**GRPO:** \<think\>If all laptops were removed, there would be 2 monitors on the desk: one being the desktop computer and one being the **laptop**. So, there would be a total of 1 monitor from the desktop computer and **1 monitor from the laptop**, making it 2 monitors. \</think\>

boxed{**2**}

**PAPO:** \<think\>I need to determine how many monitors would be present if all laptops were removed. Currently, there are **two monitors visible on the desk**. **A laptop would typically have two monitors if portable**, but here, it only seems to have one. Therefore, removing the laptops **would not change** the number of monitors, as the only other devices on the desk are monitors.\</think\>

boxed{**2**}

**CFPO:** \<think\>There is **one laptop and one desktop monitor visible** in the image. If all laptops are removed, then **there would be only one monitor in the setup**. Therefore, the answer is 1 monitor.\</think\>

boxed{**1**}

*Figure 17.* Case: Monitor Counting (Inertia)

**Question:** Which direction should I go if I want to go to Forty Foot Rd?",
**Ground Truth:** right

**GRPO:** \<think\>On the green traffic sign, the direction is clearly indicated to the right. Here's the segmentation for better understanding: - Text: \"Bustard Rd\" - Sign Type: \"[Arrows]\" - Arrow Points: Left. - Text: \"Forty Foot Rd\". **This means the correct direction to reach Forty Foot Rd is to the left along Bustard Rd.** Hence, when crossing this intersection, to go towards \"Forty Foot Rd,\" **follow the right-hand rule** and turn left.\</think\>

boxed{**left**}

**PAPO:** \<think\>I need to follow the traffic signs and directions clearly visible at the intersection. There is **a green traffic light on Bustard Rd and a red light on Forty Foot Rd**. The sign indicates that **to get to Forty Foot Rd, I need to turn right because the light is green**. There's also a green sign pointing to Bustard Rd and Forty Foot Rd, confirming the direction to take.\</think\>

boxed{**left**}

**CFPO:** \<think\>The traffic sign at the intersection clearly shows directions to two roadways: Bustard Rd and Forty Foot Rd. **The left arrow indicates the direction to Bustard Rd, which is straight. The right arrow shows the direction to Forty Foot Rd, which is straight as well**. Therefore, to go to Forty Foot Rd, you should follow the right arrow which points to the right.\</think\>

boxed{**right**}

*Figure 18.* Case: Traffic Sign Interpretation (Inertia)

# D. Performance Comparisons on Newer Architectures and Larger Scales

## D.1. Performance Comparison using Qwen3-VL-2B-Thinking

*Table 5.* Performance comparison using Qwen3-VL-2B-Thinking. Qwen3-VL introduces substantial architectural changes over Qwen2.5-VL including DeepStack and Interleaved-MRoPE (Bai et al., 2025a). We leveraged the same setup/benchmarks as PAPO and the same CFPO$_G$ hyperparameters, with the GRPO/PAPO$_G$ performance metrics directly retrieved from the original PAPO study (Wang et al., 2025b). Our proposed method consistently outperforms the baselines, remaining effective on newer architectures.

| Method | Geo3k | MathVista | We-Math | MMK12 | MathVerse | LogicVista | Counting | MMMU-Pro | MathVerse-V | Average |
|--------|-------|-----------|---------|-------|-----------|------------|----------|----------|-------------|---------|
| **GRPO Baselines** | | | | | | | | | | |
| GRPO | 39.29 | 53.58 | 57.12 | 47.71 | 47.98 | 29.84 | 80.13 | 20.51 | 45.41 | 46.84 |
| PAPO$_G$ | 41.08 | 56.08 | 59.17 | 48.57 | 51.89 | 32.83 | 80.63 | 23.42 | 50.05 | 49.30 |
| **CFPO$_G$** | **44.80** | **66.96** | **68.25** | 48.04 | **61.92** | **44.45** | **82.50** | **27.96** | **58.43** | **55.92** |

## D.2. Performance Comparison using Qwen2.5-VL-7B

*Table 6.* Performance comparison using Qwen2.5-VL-7B with the same hyperparameters as CFPO$_G$ ($\gamma$=0.02 + No Ent). GRPO performance metrics are directly retrieved from the original PAPO study (Wang et al., 2025b). CFPO$_G$ improves the average accuracy from 62.30 to 63.43, outperforming PAPO$_G$ on 7/10 benchmarks. Note that these gains are achieved without additional tuning.

| Method | RealWorld-Centric Reasoning | | | | | Mathematic-Centric Reasoning | | | | | Overall |
| | C-VQA-Real | MARS-Bench | POPE | TextVQA | MMMU-Pro(V) | Geo3k | We-Math | MMk12 | MathVerse | LogicVista | AVG |
|---|---|---|---|---|---|---|---|---|---|---|---|
| | | | | | GRPO Baselines | | | | | | |
| GRPO | - | - | - | - | 35.17 | 40.18 | 68.12 | 72.26 | 66.51 | 45.62 | - |
| PAPO$_G$ | 69.76 | 54.59 | 87.70 | 81.83 | 36.76 | 39.98 | 66.55 | 72.35 | 68.50 | 44.98 | 62.30 |
| **CFPO$_G$** | **70.72** | **56.28** | **88.41** | 81.55 | **37.23** | **43.18** | **68.79** | 71.61 | 68.47 | **48.07** | **63.43** |

# E. Training Dynamics Analysis for Regularization Settings

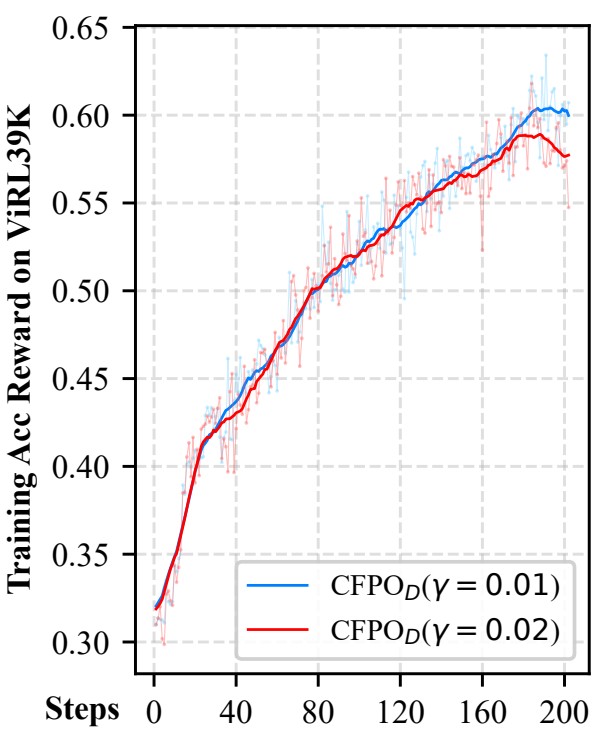 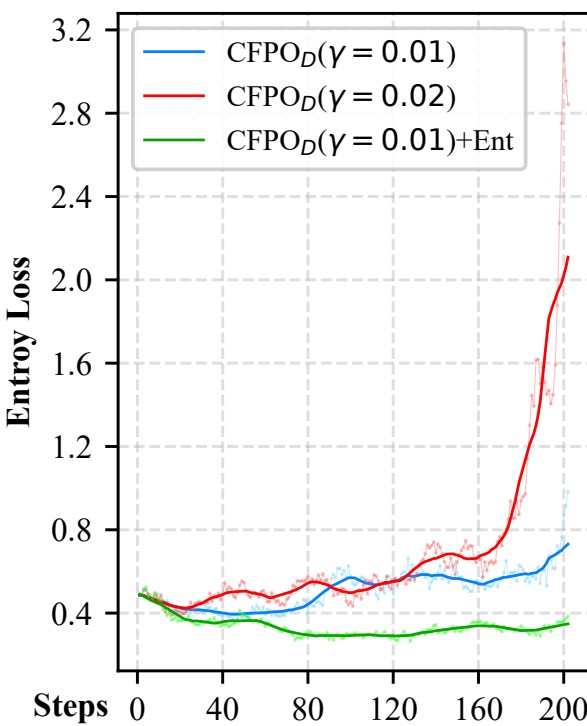

*(a)* The comparison reveals that a stronger regularization ($\gamma = 0.02$) impedes the policy's ability to optimize the primary reward in the unconstrained DAPO setting, resulting in a performance drop during final training stages (red soild line). In contrast, the milder constraint ($\gamma = 0.01$) maintains a steady upward trajectory, balancing causal intervention with task performance.

*(b)* The sharp rise in entropy loss for $\gamma = 0.02$ (red soild line) indicates severe *mode collapse*, where the policy loses exploration capability due to excessive counterfactual penalty. The inclusion of the Entropy term (green solid line) effectively stabilizes the distribution, preventing collapse and ensuring robust exploration throughout training.

*Figure 19.* Training Dynamics Analysis on Regularization Coefficients and Entropy Loss for $CFPO_D$. These dynamics confirm that DAPO, lacking the reference KL anchor, requires a delicate balance of regularization strength ($\gamma = 0.01$) and entropy maximization (+Ent) to avoid optimization instability.

