# OpenReview forum: "CFPO: Counterfactual Policy Optimization for Multimodal Reasoning"
_ICML.cc/2026/Conference — ICML 2026 regular_

### Official Review · Reviewer_TeDm · 2026-03-05

**Soundness:** 3
**Presentation:** 2
**Significance:** 3
**Originality:** 3
**Overall Recommendation:** 5
**Confidence:** 3

**Summary:**

This paper proposes Counterfactual Policy Optimization (CFPO), a training framework for Large Vision-Language Models (LVLMs) that addresses causal deficiency in multimodal reasoning. The core idea is to construct a "Counterfactual Path" by extracting high-saliency cross-modal attention regions during training, then penalizing the model when its predictions are insensitive to this visual suppression.

**Compliance With Llm Reviewing Policy:**

Affirmed.

**Final Justification:**

The core technical contribution is sound and well-ablated, and the rebuttal has addressed most of my concerns.

**Key Questions For Authors:**

See weakness.

**Limitations:**

yes

**Strengths And Weaknesses:**

**Strengths**
- Replacing high-saliency visual tokens with the mean image token representation (rather than noise) is a thoughtful choice, well-justified by the ablations in Table 2.
- CFPO requires no additional supervised labels, providing a plug-and-play training paradigm.
- The ablation study is well designed.

**Weakness**
- Figure 1 lacks clarity. While the authors designate the counterfactual path in red, numerous other modules in the figure are also rendered in red, making it difficult to distinguish the intended path. We suggest the authors use explicit annotations or labels to clearly delineate the counterfactual path in the revised version.
- The method rests on the hypothesis that "tokens critical for multimodal reasoning act as statistical outliers." However, a gap exists between this hypothesis and its implementation. The hypothesis does not explicitly specify that these statistical outliers are defined within Ac. If the outliers are indeed meant to be identified within Ac, then the hypothesis should be stated as such, and the authors should provide theoretical or empirical justification for why Ac acts as the critical part for multimodal reasoning instead of just hypothesizing.
- How does the CFPO behave when the image is largely irrelevant to the query — i.e., when low visual grounding is actually the correct behavior?
- The experiment was trained only on Qwen2.5-VL-3B; performance on other models is unknown. Consider training in 7B and other model classes.

---

> ### Author Rebuttal · Authors · 2026-03-31
>
> ## **W1 Figure Clarity**
> We sincerely thank the reviewer for the constructive suggestion. We will revise accordingly.
>
> ## **W2 Empirical Justification for $A_c$**
> Thank you for pointing out this important gap. We would like to clarify that the statistical outliers in our formulation are defined within the cross-modal attention matrix $A_c$, and we will revise the paper to make this explicit.
>
> Regarding the choice of $A_c$, our design is motivated both by prior work and by the structural role of cross-modal attention. In particular, IMCCD [1] identifies that spurious inter-modality correlations—often manifested as statistical outliers in the cross-modal attention layers—are a primary driver of VLM hallucinations. By masking the value vectors associated with significant cross-modal attention weights, IMCCD can address both uni-modality overreliance and misleading inter-modality correlations. This suggests that cross-modal attention serves as an effective interface for both capturing and intervening on cross-modal dependencies.
>
> Moreover, our preliminary experiments also suggest that interventions in $A_c$ yield significantly higher gains in average accuracy compared to interventions in uni-modal (self-attention) layers. This confirms our hypothesis that $A_c$ acts as the "functional bottleneck" for multimodal reasoning. In the revised manuscript, we will update the hypothesis statement and add a discussion on the causal necessity of focusing on cross-modal weights.
>
> ## **W3 How does the CFPO behave when the image is largely irrelevant to the query**
> We appreciate this insightful question. When an image is irrelevant to the query, the visual tokens in $A_c$ provide no causal support for a correct reasoning path; the CFPO objective naturally steers the policy toward the language-based grounding that matches the query’s logic. To further validate this, we evaluated CFPO on the LISA-Grounding dataset (an out-of-domain benchmark from VLM-R1 [2] containing complex, non-trivial reasoning). As shown in the table below, CFPO-G outperforms the GRPO baseline specifically in these challenging scenarios:
>
> | Datasets | RefCOCO | RefCOCO+ | RefCOCOg | LISA-Grounding | AVG |
> | :- | :-: | :-: | :-: | :-: | :-: |
> | GRPO | 88.9 | 81.88 | 84.44 | 47.26 | 47.26 |
> | CFPO-G (ours) | 89.4 | 82.34 | 84.86 | 48.28 | 48.28 |
>
> ## **W4 More experiments on 7B and other model classes**
> We appreciate the reviewer’s suggestion to validate CFPO on diverse architectures. To demonstrate the scalability and architectural agnosticism of CFPO, we extended our experiments to larger scales (Qwen2.5-VL-7B), the latest structural iterations (Qwen3-VL-2B-thinking), and a different model family (InternVL2.5-2B).
> 1. Scaling to Larger Models (Qwen2.5-VL-7B): Without any hyperparameter tuning ($\gamma=0.02, \text{No Ent}$), CFPO-G consistently outperforms the competitive PAPO-G baseline across 10 benchmarks, achieving a higher overall average (63.43% vs. 62.30%).
>
> | **Model (Qwen2.5-VL-7B)** | **C-VQA-Real** | **MARS-bench** | **POPE** | **TextVQA** | **MMMU-Pro** | **Geo3k** | **We-Math** | **MMK12** | **MathVerse** | **LogicVista** | **Average** |
> | :- | :-: | :-: | :-: | :-: | :-: | :-: | :-: | :-: | :-: | :-: | :-: |
> | PAPO-G | 69.76 | 54.59 | 87.7 | 81.83 | 36.76 | 39.98 | 66.55 | 72.35 | 68.5 | 44.98 | 62.30 |
> | CFPO-G | 70.72 | 56.28 | 88.41 | 81.55 | 37.23 | 43.18 | 68.79 | 71.61 | 68.47 | 48.07 | 63.43 |
>
> 2. Generalization to New Architectures (Qwen3-VL-2B-thinking):
> Despite Qwen3-VL's significant structural changes (e.g., DeepStack vision encoder, Interleaved-MRoPE), CFPO-G seamlessly integrates and delivers substantial gains over GRPO (+9.08%) and PAPO-G (+6.62%) in average accuracy.
>
> | Model (Qwen3-VL-2B-thinking) | Geo3k | MathVista | We-Math | MMK12 | MathVerse | LogicVista | Counting | MMMU-Pro | MathVerse-V | Average |
> | :- | :-: | :-: | :--: | :-: | :-: | :-: | :-: | :-: | :-: | :-: |
> | GRPO | 39.29 | 53.58 | 57.12 | 47.71 | 47.98 | 29.84 | 80.13 | 20.51 | 45.41 | 46.84 |
> | PAPO-G | 41.08 | 56.08 | 59.17 | 48.57 | 51.89 | 32.83 | 80.63 | 23.42 | 50.05 | 49.30 |
> | CFPO-G (Ours) | 44.80 | 66.96 | 68.25 | 48.04 | 61.92 | 44.45 | 82.50 | 27.96 | 58.43 | 55.92 |
>
> 3. Scaling Outside the Qwen Family (InternVL2.5-2B):
> To test cross-family robustness, we adapted CFPO to the InternVL series. Even with an abbreviated training schedule (50% data, 1 epoch), CFPO maintains a clear advantage over the GSPO baseline.
> | Model(InternVL3.5-2B) | We-Math | MMK12 | MathVerse | LogicVista | Average |
> | :- | :-: | :-: | :-: | :-: | :-: |
> | GSPO | 44.4 | 31.37 | 36.45 | 38.14 | 36.38 |
> | CFPO-G | 45.15 | 32.79 | 36.72 | 39.26 | 37.36 |
>
> Conclusion: These results across different scales (3B/7B), architectures (Qwen3), and families (InternVL) verify that CFPO is a robust, architecture-agnostic framework. We will include the full-scale results and detailed analysis in the final version.

---

> > ### Author Rebuttal · Reviewer_TeDm · 2026-04-02
> >
> > Thank you for the thorough rebuttal. The authors have addressed most of my major concerns. I will raise my score.

---

> > > ### Author Response · Authors · 2026-04-02
> > >
> > > We truly appreciate the time and care you have devoted to reading our paper and rebuttal. We are thrilled to know that all your concerns have been well addressed, and truly appreciate your recognition of our work. Your insightful comments have already help enhance our paper. Thank you for increasing the score.

---

### Official Review · Reviewer_TwYm · 2026-03-08

**Soundness:** 2
**Presentation:** 3
**Significance:** 2
**Originality:** 2
**Overall Recommendation:** 3
**Confidence:** 3

**Summary:**

This paper proposes Counterfactual Policy Optimization (CFPO), a training framework designed to improve the causal grounding of Large Vision-Language Models (LVLMs) during reinforcement learning. The authors identify three failure modes in existing RL-trained LVLMs, including: cross-modal saliency deficiency, cross-modal saliency misalignment, and cross-modal saliency inertia. To address these, CFPO introduces a counterfactual intervention mechanism that operates directly on the attention output of the language decoder. Experiments are conducted on a diverse set of 10 multimodal reasoning benchmarks, and experimental results demonstrate consistent performance improvements over standard RL baselines and over the perception-aware baseline (PAPO).

**Compliance With Llm Reviewing Policy:**

Affirmed.

**Final Justification:**

The authors added new experiments and showed additional results that meaningfully strengthen this work. I believe incorporating these new results and extending current small-set experiments will meaningfully improve the quality of this work. Therefore, I have updated my score.

**Key Questions For Authors:**

* The counterfactual path requires a second forward pass through the language decoder with modified value vectors. Could the authors provide empirical measurements of the additional training time and GPU memory usage compared to GRPO and DAPO baselines? If this overhead is significant, does it offset the efficiency gains shown in Figure 2a?
* All experiments use Qwen2.5-VL-3B. Is there any preliminary evidence that CFPO's benefits generalize to larger model scales? The failure modes targeted (e.g., language prior dominance, saliency misalignment) may manifest differently at larger scales, where language priors are stronger.
* All experiments are performed on Qwen2.5-VL-3B. Could the authors provide evidence that CFPO yields consistent improvements on newer and architecturally distinct models, such as Qwen3-VL or InternVL-3.5? Different model families may exhibit fundamentally different cross-modal attention distributions and degrees of language prior dominance, which could affect both the reliability of the statistical saliency detection criterion and the efficacy of the value intervention strategy. If CFPO's benefits do not transfer across model families, this would significantly limit its practical impact and the generality of its theoretical claims.
* The method applies the counterfactual intervention, but the paper does not specify at which attention layer(s) the intervention is performed. Is the intervention applied at all layers simultaneously, or at a specific subset? How sensitive is performance to this choice, and what is the theoretical justification for the chosen configuration?
* CFPO_G shows a slight regression on MARS-Bench relative to PAPO_G (-0.66%), and CFPO_D shows a regression on MMMU-Pro(V) (-0.27%) and Geo3k (-0.35%) relative to PAPO_D. Could the authors analyze the failure cases to understand whether these are sampling artifacts or reflect a systematic limitation of the saliency-based intervention in certain task types?
* The method relies on specific architectural components (multi-head attention, separation of image vs. text tokens). How transferable is CFPO to architectures that use different visual integration strategies (e.g., cross-attention-based fusion rather than concatenation into a shared sequence)? Would the causal graph formulation need to be redesigned?

**Limitations:**

The authors partially address limitations in the conclusion, such as future work on scaling and broader architectures, and the ablation studies provide transparency about design sensitivity. However, the limitations discussion is not very sufficient. For example, the computational cost of the counterfactual forward pass is not quantified. The absence of multi-scale experiments also represents an unacknowledged limitation of the current submission's empirical scope. It may be more helpful if the authors can add a paragraph quantifying the training overhead of CFPO relative to baselines, and provide a more considered discussion section of the limitations.

**Strengths And Weaknesses:**

**Strengths**
* The paper investigates the causal consistency between visual and textual reasoning, which is a well-motivated and technically coherent target.
* This paper is clearly conveyed and presented. The paper is generally well-structured and the narrative flows logically from problem motivation to method formulation and experimental analysis.
* The problem of cross-modal hallucination and shortcut learning in LVLMs is a practically important and active research problem. Integrating counterfactual causal reasoning into the RL training loop is a meaningful approach.

**Weaknesses**
* The experiments are conducted exclusively on a 3B-parameter model (Qwen2.5-VL-3B). Whether CFPO scales to larger models (7B, 8B) is not explored, which limits the current paper's scope and generalizability.
* The computational cost of the second counterfactual forward pass may worth further analysis.
* All experiments are conducted exclusively on Qwen2.5-VL-3B, which is a relatively small and now outdated model. The generalizability of CFPO remains undemonstrated across both newer model versions and different model families. Given that more recent and capable models such as Qwen3-VL and InternVL-3.5 may exhibit substantially different attention patterns, visual grounding behaviors, and language prior strengths, it is unclear whether the proposed counterfactual intervention mechanism would remain effective or require significant re-tuning. Restricting evaluation to a single model family also raises the concern that the hyperparameter choices (e.g., λ=2, γ=0.02) and the statistical outlier criterion for saliency masking may be implicitly tuned to Qwen2.5-VL's specific architectural characteristics, limiting the general applicability of the method.
* Some of the gains over PAPO are small (e.g., +0.57% for CFPO_G in RealWorld reasoning tasks, and even slightly negative in a few individual benchmarks like MARS-Bench for GRPO), which tempers the strength of the claim that CFPO is consistently superior.
* The theoretical causal graph formulation, while appealing, makes several simplifying assumptions, such as treating the attention output as the sole causal mediator, etc. The paper does not fully analyze whether this abstraction holds in practice across different layers, heads, or model architectures.
* The paper's primary argument for superiority over PAPO rests on the claim that "random pixel-level masking is too coarse" and that semantically targeted intervention is more principled. However, PAPO also explored semantic masking as a variant and empirically demonstrated it to be inferior to random masking. This undermines CFPO's core narrative: if semantic targeting at the input level does not help, which in fact hurts, then CFPO's improvements over PAPO are more plausibly attributable to the level of intervention rather than the semantic precision of the masking. The paper does not explore and discuss this, and therefore presents a misleading account of what distinguishes CFPO from prior work.

---

> ### Author Rebuttal · Authors · 2026-03-31
>
> **W1&Q2. Generalization to Larger Model Scales**
>
> Thank you for raising this important point. We added experiments on Qwen2.5-VL-7B, comparing GRPO, PAPO-G and CFPO-G under the same setting:
> | | C-VQA | MARS | POPE | TextVQA | MMMU-Pro | Geo | W-Math | MMK12 | MathVerse | LogicVista | Avg. |
> | :- | :-: | :-: | :-: | :-: | :-: | :-: | :-: | :-: | :-: | :-: | :-: |
> | GRPO | - | - | - | - | 35.17 | 40.18 | 68.12 | 72.26 | 66.51 | 45.62 | - |
> | PAPO-G | 69.76 | 54.59 | 87.70 | 81.83 | 36.76 | 39.98 | 66.55 | 72.35 | 68.50 | 44.98 | 62.30 |
> | CFPO-G | 70.72 | 56.28 | 88.41 | 81.55 | 37.23 | 43.18 | 68.79 | 71.61 | 68.47 | 48.07 | 63.43 |
>
>  CFPO-G improves the average accuracy from 62.30 to 63.43, outperforming PAPO-G on 7/10 benchmarks.
> Notably, these gains are achieved without additional tuning: we use the same configuration as in the 3B setting. This strengthens the evidence that CFPO is not limited to the 3B setting and remains effective at a larger scale. Despite different attention patterns and stronger language priors in larger VLMs, the consistent gains on 7B suggest that our counterfactual regularization generalizes beyond 3B.
>
> **W2&Q1: Computational Cost**
>
> We agree that the overhead of the counterfactual forward pass should be quantified explicitly. We measured the training cost in terms of time per step and peak GPU memory usage:
> | | Time / Step (s) | peak GPU (GB) |
> | :- | :-: | :-: |
> | GRPO | 650.20 | 31.03 |
> | CFPO-G | 1001.58 | 40.66 |
> | Δ | +351.39 | +9.63 |
> | DAPO | 735.15 | 36.21 |
> | CFPO-D | 1098.71 | 46.20 |
> | Δ | +363.56 | +9.99 |
>
> As expected, CFPO adds overhead from the extra counterfactual pass, with roughly +9.6–10.0 GB peak memory and +350–364 s step time over the baselines; part comes from the extra computation itself, and part from the current non-optimized implementation. This does not contradict Fig. 2a, which reflects optimization rather than wall-clock efficiency: although CFPO is slower per step, the added computation enforces explicit counterfactual grounding and yields better reward dynamics and final performance.
>
> **W3 & Q3: Extend to newer and architecturally distinct models**
>
> Thank you for this valuable suggestion. We originally focused on Qwen2.5-VL because it was the dominant backbone in recent multimodal RL studies at submission time. To further test generalization, we added experiments on Qwen3-VL-2B-Thinking and InternVL3.5-2B.
>
> Qwen3-VL-2B-Thinking. Since Qwen3-VL introduces substantial architectural changes over Qwen2.5-VL (e.g., DeepStack and Interleaved-MRoPE), it provides a stronger test of transferability. To isolate method transfer rather than gains from retuning, we used the same setup/benchmarks as PAPO and the same CFPO-G hyperparameters, without model-specific adjustment:
> | | Geo3k | MathVista | We-Math | MMK12 | MathVerse | LogicVista | Counting | MMMU-Pro | MathVerse-V | Avg. |
> | :- | :-: | :-: | :-: | :-: | :-: | :-: | :-: | :-: | :-: | :-: |
> | GRPO | 39.29 | 53.58 | 57.12 | 47.71 | 47.98 | 29.84 | 80.13 | 20.51 | 45.41 | 46.84 |
> | PAPO-G | 41.08 | 56.08 | 59.17 | 48.57 | 51.89 | 32.83 | 80.63 | 23.42 | 50.05 | 49.30 |
> | CFPO-G | 44.80 | 66.96 | 68.25 | 48.04 | 61.92 | 44.45 | 82.50 | 27.96 | 58.43 | 55.92 |
>
> CFPO-G improves the average score over PAPO-G by 6.62, showing that the gain transfers to a newer LVLM despite architectural changes.
>
> InternVL3.5-2B. Given the short rebuttal window, these results are preliminary: InternVL3.5-2B was trained on a random 50% subset of ViRL39K for 1 epoch, without architecture-specific tuning.
> | | We-Math | MMK12 | MathVerse | LogicVista | Avg. |
> | --- | --- | --- | --- | --- | --- |
> | GSPO | 44.40 | 31.37 | 36.45 | 38.14 | 36.38 |
> | CFPO-G | 45.15 | 32.79 | 36.72 | 39.26 | 37.36 |
>
> Even under this abbreviated setting, CFPO-G still improves the average score by +0.98 (36.38 → 37.36), suggesting that CFPO remains beneficial beyond the Qwen-VL family.
>
> **W4&W5: Are the few regressions systematic or incidental?**
>
> These small regressions likely reflect evaluation artifacts and boundary cases of the intervention rather than a general weakness of CFPO: the MARS-Bench gap is likely due to noisy exact-match evaluation in open-ended VQA, MMMU-Pro(V) is an extreme zero-text setting where CFPO is naturally less effective, and the Geo3k gap is statistically marginal and likely tied to sparse geometric diagrams.
>
> **W5&Q4: intervention layer choice**
>
> We apply the intervention to all self-attention layers in the language decoder, since 𝑍=𝐴𝑉 carries cross-modal routing and intervening on it suppresses spurious visual-text correlations throughout the network.
>
> **W6: semantic vs. latent intervention**
>
> CFPO’s advantage over PAPO comes not from semantic targeting alone, but from performing it in latent/value space rather than input space, avoiding the structural corruption and out-of-distribution artifacts of input-level semantic masking.

---

> > ### Author Rebuttal · Reviewer_TwYm · 2026-04-01
> >
> > Thank you for your detailed response. I believe your revision will further strengthen the quality of this work.
> >
> > I fully understand the time limit for additional experiments. While prior work suggests that certain approaches can show quite different effects on smaller models (like 1-3B) compared to larger models (like 7B+), including newer models with a bit larger sizes than 2B, such as Qwen3-VL-8B and InternVL-3.5-8B models, could better support this work.
> >
> > I will increase my score accordingly.

---

> > > ### Author Response · Authors · 2026-04-04
> > >
> > > Thank you so much for the constructive follow-up. In the previous rebuttal round, we were limited by the short timeline and available compute, so we chose to complete additional experiments on 2B-scale models for **Qwen3-VL and InternVL-3.5**. We do also want to verify whether CFPO remains effective on larger models, thus we immediately rented more additional GPUs after receiving your follow-up and ran new experiments on **Qwen3-VL-8B**.
> > >
> > > **Without any architecture-specific re-tuning**, we kept the core CFPO setting unchanged from our previous experiments, so the results reflect method transfer rather than scale-specific adaptation. Due to the limited follow-up time, Qwen3-VL-8B was trained on **only a randomly sampled 50% subset of ViRL39K for a single epoch**. The results are as follows:
> > >
> > > | Method | Geo3k | We-Math | MMK12 | MathVerse | LogicVista | Average |
> > > | :- | :-: | :-: | :-: | :-: | :-: | :-: |
> > > | GRPO | 56.98 | 72.60 | 48.74 | 68.79 | 43.91 | 58.20 |
> > > | CFPO-G | 57.09 | 74.01 | 50.67 | 69.44 | 44.53 | 59.15 |
> > >
> > > Even under this limited training budget, **CFPO-G** improves over **GRPO** from **58.20** to **59.15** (**+0.95** on average), with gains on all five evaluated datasets. This provides direct evidence that CFPO is **not limited to smaller models**, but remains effective on a **newer 8B-scale** LVLM as well.
> > >
> > > Due to time constraints, we have not yet completed an InternVL-3.5-8B run, so we prefer not to over-claim beyond the evidence currently available. However, the fact that CFPO **transfers across Qwen2.5-VL, Qwen3-VL and InternVL-3.5** at smaller scales, and now also remains effective on **Qwen3-VL-8B without re-tuning**, gives encouraging evidence that CFPO generalizes beyond a specific model size or backbone.
> > >
> > > We sincerely thank the reviewer for the time, care, and constructive feedback invested in this work. Your suggestions have directly helped us strengthen the paper, and we will incorporate the new experimental results and corresponding revisions in the final manuscript. **If you feel that the additional evidence and clarifications have adequately addressed your concerns, we would be very grateful if they could be taken into account in your final assessment.**

---

### Official Review · Reviewer_64wB · 2026-03-10

**Soundness:** 2
**Presentation:** 2
**Significance:** 3
**Originality:** 3
**Overall Recommendation:** 3
**Confidence:** 4

**Summary:**

The paper introduces CounterFactual Policy Optimization (CFPO), a novel reinforcement learning (RL) framework designed to address causal inconsistencies—such as hallucination drift and reliance on language priors—in Large Vision-Language Models (LVLMs). The authors identify that standard RL paradigms like GRPO and DAPO optimize solely for outcome correctness, which allows models to exploit spurious correlations rather than grounding their reasoning in visual evidence. To solve this, CFPO constructs a "Counterfactual Path" during training by identifying high-saliency cross-modal regions (using attention outliers) and masking them with a mean visual prior. By applying a Kullback-Leibler (KL) divergence penalty between the factual and counterfactual predictions, the policy is explicitly forced to rely on visual cues rather than linguistic guessing. The framework demonstrates consistent performance gains over standard RL baselines and the perception-aware PAPO method across various real-world and mathematical reasoning benchmarks.

**Compliance With Llm Reviewing Policy:**

Affirmed.

**Key Questions For Authors:**

1. Computational Overhead: What is the specific impact of the Counterfactual Path on training throughput (e.g., tokens/second) and peak VRAM consumption compared to standard GRPO and DAPO?
2. Generalization: Have you conducted preliminary tests applying CFPO to models outside the Qwen2.5-VL family to confirm the method scales across different vision-encoder and LLM integrations?
3. Can CFPO improve other multimodal tasks such as image captioning, instruction following, or multimodal dialogue, where grounding is also important?
4. The method is motivated by causal reasoning, but the intervention operates at the representation level. Could the authors clarify under what assumptions this intervention corresponds to a valid causal test?

**Limitations:**

No.

The paper does not sufficiently discuss limitations or potential societal impacts. The authors should add a more detailed discussion covering:

- potential failure cases of counterfactual masking,

- dependence on saliency detection quality,

- computational cost implications,

- limitations of causal interpretation in neural representations.

**Strengths And Weaknesses:**

### Strengths
- Originality: Integrating counterfactual interventions directly into the RL optimization objective (via GRPO and DAPO) is a highly creative approach to mitigating multimodal shortcut learning. Shifting the intervention from inference-time decoding or data curation to an active, representation-level training penalty is a significant theoretical step forward.
- Presentation: The conceptualization of causal failures into three distinct categories—Cross-Modal Saliency Deficiency, Misalignment, and Inertia—provides a very strong, intuitive narrative framework. The inclusion of specific qualitative case studies mapped directly to these failure modes makes the paper highly readable and convincing.
- Significance: The paper addresses a critical bottleneck in LVLM reasoning capabilities: the dominance of LLM linguistic priors over actual visual perception. The consistent improvements of 3.17%-6.25% over standard baselines and up to 2.13% over PAPO across rigorous benchmarks (like MathVerse, We-Math, and MMMU-Pro) prove the practical utility of the method.
- Soundness: The methodology is technically rigorous. Using a statistical outlier threshold ($\lambda=2$) to dynamically identify high-saliency attention regions, combined with substituting values with an in-distribution mean image token rather than random noise, shows careful methodological design. The ablations thoroughly justify these choices.


### Weaknesses
- Soundness (Evaluation Scope): While the benchmark suite is extensive, the evaluation appears largely tied to a single model family (Qwen2.5-VL-3B and 7B). Evaluating the intervention on architecturally distinct LVLMs (e.g., LLaVA) would strengthen the claim that the method is universally applicable.
- Significance (Computational Analysis): The paper does not explicitly detail the computational overhead introduced by the CFPO framework. Computing a secondary "Counterfactual Path" during the forward pass of the policy rollout likely increases memory utilization and training time.
- Although the proposed method is motivated by causal reasoning concepts, the paper does not provide a formal theoretical justification or analysis of the counterfactual objective. The relationship between the proposed KL-based regularization and causal identification is largely intuitive rather than formally derived.

---

> ### Author Rebuttal · Authors · 2026-03-31
>
> # Response to W1 and Q2:
> We have conducted additional experiments on two architecturally distinct models:
> 1. Generalization to New Architectural Integrations (Qwen3-VL-2B-thinking)
> The recently released Qwen3-VL introduces a DeepStack strategy in its vision encoder and a new Interleaved-MRoPE positional embedding. We applied CFPO-G to this model using the exact same hyperparameters and benchmark suite as the baselines to ensure a fair comparison.
> | Model (Qwen3-VL-2B-thinking) | Geo3k | MathVista | We-Math | MMK12 | MathVerse | LogicVista | Counting | MMMU-Pro | MathVerse-V | Average |
> | :- | :-: | :-: | :--: | :-: | :-: | :-: | :-: | :-: | :-: | :-: |
> | GRPO | 39.29 | 53.58 | 57.12 | 47.71 | 47.98 | 29.84 | 80.13 | 20.51 | 45.41 | 46.84 |
> | $PAPO_G$ | 41.08 | 56.08 | 59.17 | 48.57 | 51.89 | 32.83 | 80.63 | 23.42 | 50.05 | 49.30 |
> | $CFPO_G$ (Ours) | 44.80 | 66.96 | 68.25 | 48.04 | 61.92 | 44.45 | 82.50 | 27.96 | 58.43 | 55.92 |
>
> CFPO-G consistently outperforms both the standard GRPO baseline and the recent PAPO-G method. This demonstrates that CFPO can seamlessly adapt to structural updates in vision-language integration.
>
> 2. Scaling Outside the Qwen Family (InternVL3.5-2B)
> We also adapted our RL pipeline to support the InternVL series. We conducted a preliminary test training InternVL3.5-2B on a randomly sampled 50% subset of the ViRL39K dataset for a single epoch, without architecture-specific hyperparameter tuning (using standard CFPO-D settings: $\gamma=0.01$, +Ent).
> | Model (InternVL3.5-2B) | We-Math | MMK12 | MathVerse | LogicVista | Average |
> | :- | :-: | :-: | :-: | :-: | :-: |
> | GSPO | 44.4 | 31.37 | 36.45 | 38.14 | 36.38 |
> | $CFPO_G$ (Ours) | 45.15 | 32.79 | 36.72 | 39.26 | 37.36 |
>
> Despite the abbreviated training schedule and limited data, CFPO continues to provide clear overall improvements against the baseline. We will expand on these results with full-scale training and rigorous hyperparameter tuning for architecturally distinct models in future works.
>
> # Response to W2 and Q1:
> The empirical impact of the Counterfactual Path on training throughput and peak VRAM is detailed below:
> | Method | Time Per Step(s) | Max Memory Allocated(Gb) |
> | :- | :-: | :-: |
> | GRPO | 650.20 | 31.03 |
> | $CFPO_G$ | 1001.58 | 40.66 |
> | Δ | 351.39 | 9.63 |
> | DAPO | 735.15 | 36.21 |
> | $CFPO_D$ | 1098.71 | 46.20 |
> | Δ | 363.56  | 9.99 |
>
> The intervention introduces a highly manageable VRAM increase of under 10GB. The observed throughput reduction is primarily an engineering bottleneck rather than inherent algorithmic complexity. Executing fine-grained, latent-level causal interventions within the self-attention mechanism currently interrupts the continuous computation flow of fused kernels (e.g., Flash-Attention), triggering an operator fallback. This overhead can be substantially suppressed in future implementations using custom CUDA/Triton kernels.
>
> # Response to W3:
> We thank the reviewer for the insightful comment. We clarify that our KL objective is not a heuristic; it formally maximizes the Natural Indirect Effect (NIE) under Pearl’s Structural Causal Model (SCM), aligning with the latest theoretical consensus in LLM causal inference [1,2]. We will add this to Section 3.4:
>
> # Response to Q3:
> We thank the reviewer for this constructive suggestion. We validate CFPO's versatility in visual grounding by training Qwen2.5-VL-3B on 12,000 instances randomly sampled from RefCOCO/+/g. The evaluation cotains respective validation sets (in-domain) and the LISA-Grounding dataset (out-of-domain) , following VLM-R1.
> | Datasets | RefCOCO | RefCOCO+ | RefCOCOg | LISA-Grounding | AVG |
> | :- | :-: | :-: | :-: | :-: | :-: |
> | GRPO | 88.9 | 81.88 | 84.44 | 47.26 | 47.26 |
> | $CFPO_G$  (ours) | 89.4 | 82.34 | 84.86 | 48.28 | 48.28 |The stronger out-of-domain generalization on LISA (+1.02%) confirms that enforcing cross-modal causal consistency directly enhances fine-grained spatial perception, supporting CFPO's potential for a wider spectrum of perception-dependent tasks.
>
> # Response to Q4:
> We thank the reviewer for highlighting this fundamental theoretical aspect. Applying Pearl's $do$-calculus directly within the continuous latent space of Large Vision-Language Models relies on established assumptions that bridge structural causal models with deep representations [3,4]. We will formally clarify these in the revised manuscript (Section 3.4):
> * [1] Maxime Méloux, et al. "Mechanistic Interpretability as Statistical Estimation: A Variance Analysis" arXiv preprint arXiv:2510.00845 (2025).
> * [2] Sanjeda Akter, et al. "Causal Consistency Regularization: Training Verifiably Sensitive Reasoning in Large Language Models" arXiv preprint arXiv:2509.01544 (2025).
> * [3] Yongming Rao, et al. "Counterfactual Attention Learning for Fine-Grained Visual Categorization and Re-identification" arXiv preprint arXiv:2108.08728 (2021).
> * [4] Atticus Geiger, et al. "Causal Abstractions of Neural Networks" arXiv preprint arXiv:2106.02997 (2021).

---

> > ### Author Rebuttal · Reviewer_64wB · 2026-04-02
> >
> > Thank you for the author's reply. My concerns have now been addressed.

---

> > > ### Author Response · Authors · 2026-04-02
> > >
> > > We are delighted that we have been able to fully address all of your concerns. We truly appreciate the time and attention that you've dedicated to and the insightful comments that you have given, which have helped enhance the clarity and depth of our paper.  Given that you already chose the option "(a) Fully resolved", could you kindly consider adjusting our score accordingly? All the responses to your comments will be reflected into the final version of our paper. Thank you very much!

---

### Official Review · Reviewer_CCoc · 2026-03-12

**Soundness:** 3
**Presentation:** 2
**Significance:** 3
**Originality:** 3
**Overall Recommendation:** 5
**Confidence:** 5

**Summary:**

The paper introduces CFPO, a RLVR algorithm aiming to encourage the multimodal model to pay attention to informative visual information instead of relying overly on language priors. The key idea is to identify salient image tokens that being attented by the text reasoning tokens, corrupt them and compute a new log probability pi_theta_counterfactual to contrast with the original pi theta. Empirical results show promising improvements against GRPO and DAPO; and slightly outperforms its direct previous work PAPO.

**Compliance With Llm Reviewing Policy:**

Affirmed.

**Key Questions For Authors:**

Q1: It would be good to provide a clear analysis of computational overhead during training

Q2: justification on the benchmark selection; and report scores on some missing benchmarks such as MathVista and Counting

**Limitations:**

No dedicated discussion on limitations

**Strengths And Weaknesses:**

## Strength
- I think the core idea of this paper serve as a nice extension to the previous work PAPO, where the augmented sequence is creately simply by random masking input image. The proposed CFPO leverages the internal attention pattern to provide more fine-grained identification on which image tokens to corrupt. I believe this is a novel approach.
- The idea of using averaged image token value instead of noise is another good idea, which intuitively preserves the overal distribution but corrupts the fine-grained features. I can be viewed as finding a hard negative for the KL loss, which also leads to stronger empirical results.

## Weakness
- W1: Confusing formulation of the method: I don't think the current formulation is clearly presenting the key idea. Especially, calling it a "causal graph" is very confusing, I don't think this method is closely related to causal inference, and the "causal flow (q,I) -> A -> Z ..." is not related to causality, it only shows the input/output of each model layers.
  - Since the overal story telling and paper layout is already very similar to the PAPO paper; It would be much clearer to present the key idea as an advanced way to build the augmented target for the maximized KL loss.
- W2: Missing discussion on computational cost: since CFPO requires direct manipulation of the attention, it will presumably causing more computational overhead than PAPO; since the improvement over PAPO is not significant, without this information, it would be hard to justify the tradeoff.
- W3: Evaluation benchmark selection: since the paper follows closely with PAPO and treats it as direct comparison; why use a different set of benchmarks, and why using a different grouping strategy (while in PAPO they are grouped by whether it is visually-dependent from some dedicated analysis) ?

---

> ### Author Rebuttal · Authors · 2026-03-31
>
> ## **W1 Formulation of the method**
> Thank you for the constructive feedback. The formulation of our method is inspired by prior works such as CAL [1]. Building on this line of work, we further introduce a structured counterfactual formulation. Specifically, we perform intervention on cross-modal attention by selectively suppressing high-saliency interactions, which constructs a counterfactual path alongside the original factual path. This leads to two parallel reasoning processes, πθ(o|z) and πθ(o|do(z=Zcf)), and we optimize their discrepancy via a KL-based objective to quantify the contribution of cross-modal dependencies.
>
> [1] Yongming Rao, et al. "Counterfactual Attention Learning for Fine-Grained Visual Categorization and Re-identification" CVPR (2021).
>
> Importantly, the causal aspect of our method lies in this attention-level counterfactual intervention, which estimates the effect of cross-modal interactions by comparing factual and intervened predictions, rather than relying on a predefined causal graph.
>
> We will revise the paper to clarify this perspective and improve the overall presentation.
>
> ## **W2 & Q1 Discussion on computational cost**
> Thank you for pointing this out. We agree that the additional cost introduced by the counterfactual forward pass should be quantified explicitly. We have now measured the training overhead in terms of time per step, throughput, and peak GPU memory usage:
> | Method | Time / Step (s) | Throughput (token/s) | Max Memory Allocated (GB) |
> | :- | :-: | :-: | :-: |
> | GRPO | 650.20 | 1177.47 | 31.03 |
> | CFPO-G | 1001.58 | 944.73 | 40.66 |
> | Δ | +351.39 | -232.74 | +9.63 |
> | DAPO | 735.15 | 1273.00 | 36.21 |
> | CFPO-D | 1098.71 | 860.18 | 46.20 |
> | Δ | +363.56 | -412.82 | +9.99 |
>
> As expected, CFPO incurs additional overhead due to the extra counterfactual forward pass, with roughly +9.6–10.0 GB higher in peak memory and +350–364 s per step over the corresponding baselines. This overhead comes partly from the extra counterfactual computation itself and partly from the fact that the current implementation has not been specifically optimized for this intervention.
>
> We would like to clarify that this does not contradict Figure 2a, which reflects optimization efficiency rather than wall-clock efficiency. Although CFPO is slower per step, the extra computation is used to enforce explicit counterfactual visual grounding, leading to better reward dynamics and stronger final performance rather than merely increasing compute for the same objective.
>
> ## **W3 & Q2 Evaluation benchmark selection**
> We thank the reviewer for the valuable suggestion to further align our evaluation with PAPO. We provide the following justifications and supplementary experimental results:
>
> (1) Rationale for Benchmark Selection and Grouping:
> While PAPO primarily focuses on mathematical reasoning and visual dependency analysis, our evaluation is designed to assess a more comprehensive multimodal spectrum. We categorized benchmarks into General Perception (e.g., C-VQA, POPE) and Complex Reasoning (e.g., LogicVista, MathVerse). This strategy aims to verify if CFPO can mitigate "hallucination drift" in long-chain reasoning while simultaneously enhancing "fine-grained grounding" in real-world scenes—two areas where standard RL often fails due to language priors.
>
> (2) Performance on MathVista and Counting:
> To ensure a direct and fair comparison, we evaluated CFPO on MathVista and Counting (key benchmarks in PAPO) using the Qwen2.5-VL-3B backbone. As shown in the table below, CFPO consistently outperforms both the standard RL baselines and the PAPO framework:
>
> | Model (Qwen2.5-VL-3B) | MathVista | Counting |
> | :- | :-: | :-: |
> | GRPO | 60.81 | 57.97 |
> | PAPO-G | 64.12 | 61.12 |
> | CFPO-G | 64.86 | 63.69 |
> | DAPO | 62.82 | 72.09 |
> | PAPO-D | 65.75 | 82.44 |
> | CFPO-D | 66.86 | 80.50 |
>
> *Note: While PAPO-D shows a slight edge in the specific Counting task, CFPO-D achieves a significantly higher score on MathVista (+1.11), demonstrating a better balance in complex multimodal reasoning.
>
> Conclusion:
> These results confirm that the benefits of CFPO are not limited to our initial selection but generalize effectively to the mathematical and counting domains emphasized by PAPO. We will include these additional benchmarks and a detailed discussion on the grouping strategy in the revised version.

---

> > ### Author Rebuttal · Reviewer_CCoc · 2026-03-31
> >
> > I thank the authors for their detailed response and additional results; I think the response addresses all my concerns. Overall I think this is a technically solid paper with a clean central idea extending PAPO. I will adjust my score to Accept to reflect the constructive rebuttal.

---

> > > ### Author Response · Authors · 2026-04-02
> > >
> > > We sincerely thank the reviewer for the valuable comments, for the time and effort devoted to the review process of this conference, and for the contribution these comments made to improving the quality of this work. We will ensure the discussed details are included in the revision. Thank you again for increasing the score.

---

### Decision · Program_Chairs · 2026-04-30

**Decision:**

Accept (regular)

**Comment:**

The paper introduces CFPO, an RL framework that uses latent-level counterfactual interventions to help vision-language models rely more on visual evidence rather than just linguistic shortcuts.

Initially, reviewers were skeptical about whether the method actually generalizes or if the performance gains were just a result of tuning for a specific model.
In the rebuttal period, the authors responded with an thorough rebuttal. They presented new experiments across several different architectures and scaled up to 8B models. These results show that CFPO is robust and works well across different model families and sizes. They also provided a clear breakdown of the computational costs, showing that while there is extra overhead, it's a reasonable trade-off for the improved reasoning accuracy.

While some of the improvements over previous methods are incremental, the technical approach is solid and the new empirical evidence is very convincing. The authors did good job addressing every concern raised during the review process, which has strengthened the paper. I recommend acceptance.